# Explicit Preference Optimization:
# No Need for an Implicit Reward Model

Xiangkun Hu [* 1]   Lemin Kong [* 2]   Tong He [1]   David Wipf [1]

## Abstract

The generated responses of large language models (LLMs) are often fine-tuned to human preferences through a process called reinforcement learning from human feedback (RLHF). As RLHF relies on a challenging training sequence, whereby a separate reward model is independently learned and then later applied to LLM policy updates, ongoing research effort has targeted more straightforward alternatives. In this regard, direct preference optimization (DPO) and its many offshoots circumvent the need for a separate reward training step. Instead, through the judicious use of a reparameterization trick that induces an *implicit* reward, DPO and related methods consolidate learning to the minimization of a single loss function. And yet despite demonstrable success in some real-world settings, we prove that DPO-based objectives are nonetheless subject to sub-optimal regularization and counter-intuitive interpolation behaviors, underappreciated artifacts of the reparameterizations upon which they are based. To this end, we introduce an *explicit* preference optimization framework termed EXPO that requires no analogous reparameterization to achieve an implicit reward. Quite differently, we merely posit intuitively-appealing regularization factors from scratch that transparently avoid the potential pitfalls of key DPO variants, provably satisfying regularization desiderata that prior methods do not. Empirical results serve to corroborate our analyses and showcase the efficacy of EXPO.

*Equal contribution  [1]Amazon Web Services  [2]The Chinese University of Hong Kong. Correspondence to: Xiangkun Hu <xkhu17@fudan.edu.cn>, Lemin Kong <1155165468@link.cuhk.edu.hk>, Tong He <htong@amazon.com>, David Wipf <davidwipf@gmail.com>.

*Proceedings of the 42nd International Conference on Machine Learning*, Vancouver, Canada. PMLR 267, 2025. Copyright 2025 by the author(s).

## 1. Introduction

Reinforcement learning from human feedback (RLHF) (Bai et al., 2022a; Ouyang et al., 2022; Stiennon et al., 2009; Ziegler et al., 2019) represents an effective means of aligning the output of powerful pre-trained large language models (LLMs) (Bubeck et al., 2023; Chang et al., 2024; Achiam et al., 2023; Zhao et al., 2023a) with human preferences (Bai et al., 2022b; Gallegos et al., 2023). On the downside though, RLHF requires a potentially-unstable multi-step process whereby first a reward model is learned using a labeled preference dataset. Subsequently, a new LLM policy is trained to maximize this reward while minimizing deviations from the original pre-trained LLM (the reference policy). To mitigate this complexity, various reparameterization schemes have recently been introduced that obviate the need for training a separate reward model. Instead, so-called direct human preference optimization (DPO) (Rafailov et al., 2024) and follow-up variants (Azar et al., 2024; Tang et al., 2024; Wang et al., 2024a; Zhao et al., 2023b) are instantiated via the minimization of a single closed-form training loss, within which an *implicit* reward function is concomitantly optimized.

Although these DPO-based alternatives to RLHF dramatically streamline the modeling pipeline, as we will detail herein, they are nonetheless saddled with underappreciated limitations of their own, such as sub-optimal regularization effects, degenerate minima, and other counter-intuitive behaviors linking back to the original reparameterizations and the viability of the implicit rewards involved. To this end, our paper is devoted to addressing these shortcomings via the introduction of intuitive new training objectives that we term EXPO, for explicit preference optimization, which rely on no such implicit rewards or associated reparameterizations. After presenting basic concepts and the details of existing preference optimization models in Section 2, the remainder of the paper devoted to our technical contributions can be distilled as follows:

- We introduce new evaluation criteria that comport with intuition regarding how a preference model ideally should behave, and yet (somewhat surprisingly) are provably *not* satisfied by a broad class of existing DPO-based approaches. In particular, we show that because

of uniform regularization effects, the minimizers of commonly-used preference optimization objectives like DPO are at times unable to *preserve* good performance in regions where the reference model is strong while *simultaneously* improving upon the reference model elsewhere (Section 3.1). Moreover, we also elucidate limitations in the ability to *interpolate* between ideal endpoints as model trade-off parameters are varied (Section 3.2).

- In Section 4 we then propose multiple new EXPO training objectives $\ell_{\text{EXPO}}$ for *explicitly* optimizing human preferences while minimizing deviations from an original reference policy. Importantly, these losses satisfy our evaluation criteria from above while requiring no reparameterization or implicit link to a separate, motivational RLHF objective. And although these $\ell_{\text{EXPO}}$ candidates depend on the unobservable ground-truth preference distribution by design, we nonetheless establish that unbiased estimates of the gradient can be directly computed to facilitate efficient SGD.

- To complement our analysis, Section 5 provides empirical verification of conditions, in a controlled environment with known ground-truth preferences, whereby DPO-based regularization and related variants converge to degenerate minimizers while $\ell_{\text{EXPO}}$ minimizers do not. We then conclude with experiments involving real-world alignment data that show EXPO outperforms DPO-based models w.r.t. response win rates.

## 2. Background

We adopt $x \sim \mathcal{D}_x$ to denote an *input prompt* $x$ drawn from some distribution $\mathcal{D}_x$. From here, conditioned on such prompts we may then generate *responses* $y$ using a pretrained reference language model/policy $\pi_{\text{ref}}(y|x)$. Moreover, given a pair of such responses $y_1 \neq y_2$, we adopt the binary indicator variable $z = \mathbb{I}[y_1 \succ y_2|y_1, y_2, x]$ to convey that $y_1$ is preferred over $y_2$ by a human evaluator when $z = 1$, or else $z = 0$ if instead $y_2 \succ y_1$. Given a population of such evaluators, we express the ground-truth human preference distribution as $p^*(z|y_1, y_2, x) = p^*(y_1 \succ y_2|y_1, y_2, x)$. And finally, we define a set of human labeled tuples drawn from a training distribution $\mathcal{D}_{tr}$ as

$$
\begin{aligned}
\{y_w, y_l, x\} \sim \mathcal{D}_{tr} &\equiv \{z, y_1, y_2, x\} \sim \mathcal{D}_{tr} \qquad (1) \\
&\equiv z \sim p^*(z|y_1, y_2, x), \{y_1, y_2\} \sim \pi_{\text{ref}}(y|x), x \sim \mathcal{D}_x,
\end{aligned}
$$

where $y_w \succ y_l$; subscripts here stand for 'win' and 'lose'.[1] In other words, each training tuple is generated by drawing $x$

---

[1]We generally assume that $y_1 \neq y_2$; however, the $y_1 = y_2$ case can nonetheless be handled by simply assigning $p^*(z|y, y, x) = 1/2$, inclusion of which does not affect the analysis that follows. In particular, such cases merely introduce an irrelevant constant into the human preference loss functions under consideration.

from $\mathcal{D}_x$, $y_1 \neq y_2$ from the reference policy $\pi_{\text{ref}}$, and finally $z$ is produced by human labelers that operate according to $p^*$. Note that per convention in prior work and ease of presentation, we will often abbreviate the preference distribution notation as $p^*(y_1 \succ y_2|y_1, y_2, x) \equiv p^*(y_1 \succ y_2|x)$ when the context is sufficiently clear. We now briefly introduce RLHF, DPO, and follow-up variants that will serve as motivation for our proposed EXPO framework.

### 2.1. RLHF

**Reward Function Estimation:** Given two candidate responses $y_1 \neq y_2$ sampled using prompt $x$, the Bradley-Terry (BT) model (Bradley & Terry, 1952) for human preferences stipulates that

$$
\begin{aligned}
p^*(y_1 \succ y_2|x) &= \frac{\exp[r^*(y_1, x)]}{\exp[r^*(y_1, x)] + \exp[r^*(y_2, x)]} \\
&= \sigma\big[r^*(y_1, x) - r^*(y_2, x)\big], \qquad (2)
\end{aligned}
$$

where $r^*(y, x)$ is a so-called latent reward model and $\sigma$ is the logistic function. Because $r^*(y, x)$ is unobservable, it is not possible to directly compute $p^*(y_1 \succ y_2|x)$; however, we can train an approximation $p_\phi(y_1 \succ y_2|x)$ defined by a parameterized proxy reward $r_\phi(y, x)$. Specifically, we can minimize the loss

$$
\begin{aligned}
\ell_{\text{BT}}(r_\phi) &:= \mathbb{E}_{\{y_w, y_l, x\} \sim \mathcal{D}_{\text{tr}}}\Big[ -\log p_\phi(y_w \succ y_l|x)\Big] \quad (3) \\
&= \mathbb{E}_{\{y_w, y_l, x\} \sim \mathcal{D}_{\text{tr}}}\Big[ -\log \sigma\big[r_\phi(y_w, x) - r_\phi(y_l, x)\big]\Big].
\end{aligned}
$$

The optimized reward $\hat{r}_\phi(y, x) := \arg\min_{r_\phi} \ell_{\text{BT}}(r_\phi) \approx r^*(y, x)$ can then be applied to fine-tuning the pre-trained reference model $\pi_{\text{ref}}(y|x)$ as described next.

**RL Fine-Tuning with an Estimated Reward:** Given the optimized reward from above, the remaining goal is to improve upon a given pre-trained LLM, or reference policy $\pi_{\text{ref}}(y|x)$, using a separate trainable model $\pi_\theta(y|x)$. This is accomplished by minimizing the RLHF loss in the form

$$
\begin{aligned}
\ell_{\text{RLHF}}(\pi_\theta, \pi_{\text{ref}}, \hat{r}_\phi, \lambda) &:= \mathbb{E}_{y \sim \pi_\theta(y|x), x \sim \mathcal{D}_x}\Big[ -\hat{r}_\phi(y, x)\Big] \\
&+ \lambda \mathbb{E}_{x \sim \mathcal{D}_x}\Big[\mathbb{KL}\big[\pi_\theta(y|x) || \pi_{\text{ref}}(y|x)\big]\Big], \qquad (4)
\end{aligned}
$$

where $\lambda > 0$ is a trade-off parameter balancing a reward maximization term and the KL divergence from the reference policy. Although not directly amenable to SGD, given an initialization $\pi_\theta = \pi_{\text{ref}}$, the loss $\ell_{\text{RLHF}}(\pi_\theta, \pi_{\text{ref}}, \hat{r}_\phi, \lambda)$ can still be optimized over $\pi_\theta$ using RL techniques (Schulman et al., 2017; Ramamurthy et al., 2022).

### 2.2. DPO

DPO (Rafailov et al., 2024) is based on the observation that, for an arbitrary reward function $r(y, x)$, the minimum

of $\ell_{\text{RLHF}}\left(\pi_\theta, \pi_{\text{ref}}, r, \lambda\right)$ w.r.t. $\pi_\theta$ can be computed in closed form as

$$\pi_r(y|x) \quad := \quad \arg\min_{\pi_\theta} \ell_{\text{RLHF}}\left(\pi_\theta, \pi_{\text{ref}}, r, \lambda\right) \qquad (5)$$

$$= \quad \frac{1}{Z(x)}\pi_{\text{ref}}(y|x)\exp\left[\frac{1}{\lambda}r(y,x)\right],$$

where $Z(x) := \sum_y \pi_{\text{ref}}(y|x)\exp\left[\frac{1}{\lambda}r(y,x)\right]$ ensures that $\pi_r(y|x)$ forms a proper distribution (Peng et al., 2019; Peters & Schaal, 2007). From here, we can invert (5) to express the reward via the policy as

$$r(y,x) = \lambda\log\frac{\pi_r(y|x)}{\pi_{\text{ref}}(y|x)} + \lambda\log Z(x). \qquad (6)$$

As no assumptions have been placed on $r$, it follows that (5) and (6) hold even for the ground-truth reward $r^*$ and the corresponding optimal policy $\pi^{**}(y|x) := \arg\min_{\pi_\theta} \ell_{\text{RLHF}}\left(\pi_\theta, \pi_{\text{ref}}, r^*, \lambda\right)$. Therefore instead of approximating $r^*(y,x)$ with $r_\phi(y,x)$ as in (2), we may approximate $\pi^{**}(y|x)$ with some $\pi_\theta(y|x)$ within (6). The resulting reparameterized BT objective then forms the DPO loss

$$\ell_{\text{DPO}}(\pi_\theta, \pi_{\text{ref}}, \lambda) \ := \ \ell_{\text{BT}}\left(\lambda\log\frac{\pi_\theta(y|x)}{\pi_{\text{ref}}(y|x)}\right)$$

$$\equiv \ \mathbb{E}_{\{y_w, y_l, x\}\sim\mathcal{D}_{\text{tr}}}\left[-\log\sigma\left(\lambda\log\frac{\pi_\theta(y_w|x)}{\pi_{\text{ref}}(y_w|x)}\right.\right.$$

$$\left.\left. - \ \lambda\log\frac{\pi_\theta(y_l|x)}{\pi_{\text{ref}}(y_l|x)}\right)\right], \quad (7)$$

where the $\lambda\log Z(x)$ term cancels out and can be omitted. And so it becomes possible to efficiently optimize (7) over $\pi_\theta$ using SGD without RL. Overall, the policy $\pi_\theta$ induces an *implicit* reward $\lambda\log\left[\pi_\theta(y|x)\pi_{\text{ref}}^{-1}(y|x)\right]$ that is optimized by minimizing the reparameterized BT model.

### 2.3. Identity Preference Optimization (IPO)

The identity preference optimization (IPO) framework (Azar et al., 2024) weaves an alternative reward function into the original RLHF loss from (4) such that efficient learning is possible as with DPO. Concretely, IPO is designed to minimize $\ell_{\text{RLHF}}\left(\pi_\theta, \pi_{\text{ref}}, r_{\text{IPO}}, \lambda\right)$ over $\pi_\theta$, where

$$r_{\text{IPO}}(y,x) := \mathbb{E}_{y'\sim\pi_{\text{ref}}(y|x)}\left[p^*(y \succ y'|x)\right]. \qquad (8)$$

Stemming from the special structure of this particular reward, $\ell_{\text{RLHF}}\left(\pi_\theta, \pi_{\text{ref}}, r_{\text{IPO}}, \lambda\right)$ can be minimized without RL. This is accomplished using an analogous reparameterization trick to DPO, leading to the final IPO loss $\ell_{\text{IPO}}(\pi_\theta, \pi_{\text{ref}}, \lambda) \ :=$

$$\mathbb{E}_{\{y_w, y_l, x\}\sim\mathcal{D}_{\text{tr}}}\left(\log\left[\frac{\pi_\theta(y_w|x)\pi_{\text{ref}}(y_l|x)}{\pi_\theta(y_l|x)\pi_{\text{ref}}(y_w|x)}\right] - \frac{1}{2\lambda}\right)^2, \quad (9)$$

which is naturally amenable to SGD like DPO. Further details regarding properties of the IPO loss from (9) are contained in Appendix E.5.

### 2.4. Flexible Quasi-Convex Generalizations

From the expressions above, it is clear that both DPO and IPO reduce to functions of $\log\left[\frac{\pi_\theta(y_w|x)\pi_{\text{ref}}(y_l|x)}{\pi_\theta(y_l|x)\pi_{\text{ref}}(y_w|x)}\right]$ and a tunable hyperparameter $\lambda$. As such, it is natural to consider extensions to broader choices in the form

$$\ell_{\text{QPO}}(\pi_\theta, \pi_{\text{ref}}, \psi, \mu, \lambda) := \qquad (10)$$

$$\mathbb{E}_{\{y_w, y_l, x\}\sim\mathcal{D}_{\text{tr}}} \ \psi\left(\mu\left[\frac{\pi_\theta(y_w|x)}{\pi_{\text{ref}}(y_w|x)}\right] - \mu\left[\frac{\pi_\theta(y_l|x)}{\pi_{\text{ref}}(y_l|x)}\right], \lambda\right),$$

where $\mu : \mathbb{R}^+ \to \mathbb{R}$ is a monotonically increasing function (which generalizes the logarithm), and the function $\psi : \mathbb{R} \times \mathbb{R}^+ \to \mathbb{R}$ influences the overall loss shape. We stipulate that $\psi$ is a differentiable *quasi-convex* function (Greenberg & Pierskalla, 1971); hence the chosen loss notation $\ell_{\text{QPO}}$ for quasi-convex preference optimization. By definition of quasi-convexity, $\psi$ monotonically increases to the right or left away from the minimum.

These specifications cover DPO and IPO as representative special cases, and include essentially all reasonable choices for a loss within this family, e.g., it is nonsensical to include multi-modal losses. The generalized preference optimization (GPO) (Tang et al., 2024) and $f$-DPO (Wang et al., 2024a) frameworks are also special cases of QPO as defined herein. With GPO, $\mu$ is a logarithm and $\psi$ is chosen as an arbitrary convex function (such as used by SLiC (Zhao et al., 2023b)). Meanwhile $f$-DPO involves $\psi(\cdot, \lambda) = -\log\sigma[\lambda(\cdot)]$ analogous to DPO but with $\mu = f'$, where $f'$ denotes the derivative of an $f$-divergence (Rubenstein et al., 2019); given that $f$ must be convex, its derivative will necessarily be monotonically increasing. In this way, the RLHF objective from (4) is still optimized via $f$-DPO, but with an $f$-divergence replacing the KL term.

While overall quite general, we will nonetheless later demonstrate that any loss in the form of (10) will unavoidably be saddled with certain limitations. See also Appendices A.1 and B for further context w.r.t. recent DPO-related work that lies outside of our present scope.

### 2.5. Online vs Offline Preference Optimization

Some recent work (Guo et al., 2025; Tajwar et al., 2024; Xu et al., 2024) has suggested that so-called *online* preference learning (as in the original RLHF that trains using samples from $\pi_\theta$) may often produce better results than *offline* learning (as in DPO which only uses fixed samples from $\pi_{\text{ref}}$). There are nonetheless justifiable reasons for still considering the latter. However, in Appendix B we provide rationale for why exploration of offline approaches is still warranted, especially when we consider methodology that extends beyond the original DPO script as is our focus.

## 3. Limitations of Existing Approaches

We now turn to formalizing previously-unexplored limitations of existing DPO-like approaches. Throughout this section we will rely on the notion of an optimal policy $\pi^*$. We intentionally do not specify by what criteria this optimality is established, as our results will hold for *any* (non-deterministic) policy such that $\pi^*(y|x) \in (0, 1)$ for all $x \in \mathcal{X}$.

### 3.1. Failure to Preserve Optimal Policies

Consider the following plausible scenario, variations of which are likely to occur (at least in varying degrees) with real-world data. Suppose the support of prompts generated by $\mathcal{D}_x$ partitions as $d_x^{good} \cup d_x^{bad}$, with $d_x^{good} \cap d_x^{bad} = \emptyset$. Furthermore, assume we have access to a reference policy $\pi_{ref}$ such that $\pi_{ref} = \pi^*$ for $x \in d_x^{good}$ and $dist[\pi_{ref}, \pi^*] \gg 0$ for $x \in d_x^{bad}$, where $dist[\cdot, \cdot]$ is an arbitrary distance measure. In other words, when evaluated w.r.t. a policy $\pi^*$ that proportionally reflects human preferences, $\pi_{ref}$ performs ideally on a subset of prompts but not on others.

This dichotomy provides a useful lens for examining certain loss function properties. In particular, we would like any policy that minimizes a candidate loss to preserve $\pi_{ref}$ for prompts $x \in d_x^{good}$, while pushing away from $\pi_{ref}$ towards $\pi^*$ for prompts $x \in d_x^{bad}$. However, because of uniform regularization effects intrinsic to QPO losses, it is not actually possible to achieve even this modest objective.

**Theorem 3.1.** *(Informal version) Given the prompt partitioning, reference policy, and optimal policy described above, define $\hat{\pi}_\theta^{QPO} := \arg\min_{\pi_\theta} \ell_{QPO}(\pi_\theta, \pi_{ref}, \psi, \mu, \lambda)$ for any fixed selection of $\{\psi, \mu, \lambda\}$. Then under relatively mild assumptions on the labeled responses in $\mathcal{D}_{tr}$, if $dist[\hat{\pi}_\theta^{QPO}, \pi^*] < dist[\pi_{ref}, \pi^*]$ for $x \in d_x^{bad}$, then $dist[\hat{\pi}_\theta^{QPO}, \pi^*] > 0$ for $x \in d_x^{good}$.*

The proof and formal version are provided in Appendix D.1, while Figure 1 provides an illustration. The somewhat unexpected implication here is that if we minimize any possible QPO loss in the form of (10) and improve the policy quality in areas where $\pi_{ref}$ performs poorly w.r.t. $\pi^*$, then it *must also be the case that performance becomes worse in areas where $\pi_{ref}$ was originally optimal*. This phenomena represents an unavoidable trade-off when we restrict ourselves to using a QPO loss, of which DPO, IPO, GPO, and $f$-DPO are all special cases inheriting the same limitation. The core issue is that these QPO losses *unselectively* apply the same regularization, starting from the same initialization point, to both good and bad cases relative to an arbitrary $\pi^*$.

**Intuition from Special Cases:** Although Theorem 3.1 is presented in a technical form to rigorously cover QPO cases more broadly, a more intuitive viewpoint can be established

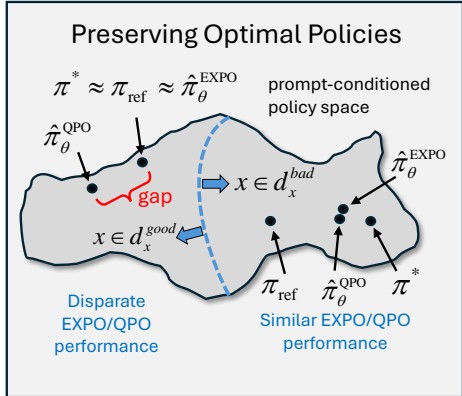

Figure 1: Preservation of optimal policies; proposed EXPO approach (Sec. 4) included for added context. The dashed blue line divides prompt space into regions where $\pi_{ref}$ performs poorly (right side, $d_x^{bad}$) from regions where it is already near optimal (left side, $d_x^{good}$). Within $d_x^{bad}$ we observe consistent movement towards $\pi^*$ as desired; however, within $d_x^{bad}$ we see that $\hat{\pi}_\theta^{QPO}$ introduces an unwanted gap from $\pi^*$ (unlike EXPO).

when we zoom in to examine DPO and IPO specifically. Both losses (i.e., (7) and (9)) depend on $\pi_\theta$ exclusively as a function of $\log\left[\frac{\pi_\theta(y_w|x)\pi_{ref}(y_l|x)}{\pi_\theta(y_l|x)\pi_{ref}(y_w|x)}\right]$, which equates to zero whenever $\pi_\theta = \pi_{ref}$, regardless of preference data. And yet by straightforward inspection, we observe that both DPO and IPO losses can always be reduced further as this log factor moves away from zero for virtually any empirical preference distribution. This forces $\pi_\theta$ away from $\pi_{ref}$, *even when $\pi_{ref} = \pi^*$*, however the latter is defined.

### 3.2. Suboptimal Interpolation

As the underlying goal shared by all approaches is to *balance* proximity to a reference policy $\pi_{ref}$ with respect for the human preference model $p^*$, a non-negative trade-off parameter $\lambda \in [a, b]$ that allows for interpolating between these competing objectives is inevitable, where $a \in \mathbb{R}$ and $b \in \mathbb{R}$ are lower and upper bounds respectively.[2] In this section we examine more closely the nature of loss function minimizers as $\lambda$ is varied, zooming in on their behavior in the limit as $\lambda \to a$ and $\lambda \to b$. To this end, we first introduce the following definitions :

**Definition 3.2.** We say that an arbitrary preference optimization loss $\ell(\pi_\theta, \pi_{ref}, \lambda)$ satisfies the **strong interpolation criteria (SIC)** if the following hold:

1. $\lim_{\lambda \to a} \arg\min_{\pi_\theta} \ell(\pi_\theta, \pi_{ref}, \lambda) = \pi^*$;

---

[2]Depending on the method, if $a = 0$ or $b = \infty$ we may replace the $\lambda$ range with an open set.

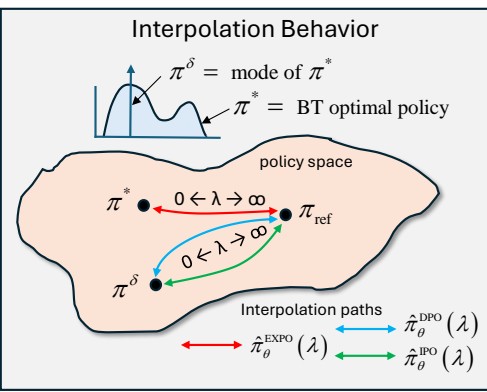

Figure 2: Interpolation illustration; a proposed EXPO variant (Sec. 4.1) is included for added context.

2. $\lim_{\lambda \to b} \arg \min_{\pi_\theta} \ell(\pi_\theta, \pi_{\text{ref}}, \lambda) = \pi_{\text{ref}}$;

3. For all other $\lambda \in (a, b)$, the optimal policy interpolates between the above two extremes.

**Definition 3.3.** For any prompt $x$ and response $y$ define[3]

$$\pi^\delta(y|x) := \arg \max_{\pi_\theta} \mathbb{E}_{y \sim \pi_\theta(y|x)} \left[ r^*(y, x) \right] \quad (11)$$

$$= \begin{cases} 1 & \text{if } y = \arg \max_{y'} \pi^*(y|x) \\ 0 & \text{otherwise.} \end{cases}$$

In this way, $\pi^\delta(y|x)$ assigns probability one to the mode of $\pi^*(y|x)$, i.e., akin to a delta function *with no generation diversity whatsoever*. We then say that a loss $\ell(\pi_\theta, \pi_{\text{ref}}, \lambda)$ satisfies the **weak interpolation criteria (WIC)** analogously to the SIC, only for the lower bound we instead require $\lim_{\lambda \to a} \arg \min_{\pi_\theta} \ell(\pi_\theta, \pi_{\text{ref}}, \lambda) = \pi^\delta$.

In summary, the only difference between these interpolation criteria is their limiting behavior w.r.t. the lower bounding $\lambda$; for the SIC we approach the optimal policy (however it is defined), while for the WIC we approach a degenerate policy with all probability mass restricted to the *mode* of the optimal policy. We remark that both the SIC and WIC cannot be simultaneously satisfied unless $\pi^*$ itself is a degenerate delta function. See also Appendix E.2 where we provide an illustrative example of why the intrinsic diversity of $\pi^*$ may be preferred over $\pi^\delta$.

We now explore how these distinctions are reflected in the behavior of DPO and IPO loss minimizers, with Figure 2 illustrating the basic concepts.

**Proposition 3.4.** *Assume preference data distributed according to $\mathcal{D}_{tr}$ from (1), and that $p^*(y_1 \succ y_2|x) \in (0, 1)$ for all responses with $\pi_{ref}(y|x) > 0$. Then the DPO loss from (7) satisfies the WIC (but not the SIC).*

---
[3]See Appendix E.1 for the derivation of the right-hand equality in (11).

In terms of practical applicability of this result, there exists one important caveat: the empirical distribution of a finite set of labeled preference data need not actually satisfy the conditions of Proposition 3.4. For example, suppose for each prompt $x \in \mathcal{D}_x$ we collect only two responses $\{y_1, y_2\}$ along with a single preference label $z$, which together produce the tuple $\{y_w, y_l, x\}$. In this scenario, which reflects certain publicly-available human preference datasets (Bai et al., 2022a; Ganguli et al., 2022), the empirical distribution of preferences will be $p^*(y_w \succ y_l|x) = 1 \notin (0, 1)$ for all $x \in \mathcal{D}_x$. Notably, Proposition 3.4 will *not* hold, and in particular, it can be easily shown that minimizers of any valid $f$-DPO loss will be *completely independent* of $\pi_{\text{ref}}$ for all $\lambda \in (0, \infty)$; in other words, *no interpolation occurs at all*; see Appendix E.3 for the derivation. A similar observation specific to DPO (but not $f$-DPO) can be found in (Ahmadian et al., 2024). The fact that DPO-based solutions may still reflect $\pi_{\text{ref}}$ in practice, and more-so as $\lambda$ increases, relates to implicit constraints and subtle regularization effects as detailed in (Kong et al., 2025).

**Proposition 3.5.** *Assume preference data distributed according to $\mathcal{D}_{tr}$ from (1). Then the IPO loss from (9) satisfies the WIC (but not the SIC).*

Comparing Proposition 3.5 with Proposition 3.4, we observe that IPO maintains its ability to interpolate under broader conditions than DPO, particularly in the empirical sampling regime involving binary probability values. That being said, neither DPO nor IPO satisfy the SIC, which motivates consideration of alternative losses that do, at least if our priority is to actually achieve the SIC (which of course may depend on the application scenario). For this purpose, it turns out that selections *beyond* the family of QPO objectives (which includes DPO, IPO, GPO, and $f$-DPO) are necessary per the following:

**Theorem 3.6.** *Assume preference data distributed according to $\mathcal{D}_{tr}$ from (1). Then no possible QPO loss from (10) will satisfy the SIC.*

## 4. Explicit Preference Optimization

Motivated by the analysis in Section 3, we now examine alternative objective functions *outside of the QPO family* adhering to the following desiderata:

1. **Perservation:** Capable of selectively preserving an optimal policy in ideal regimes, while *simultaneously* improving the policy in regions of poor performance (from Section 3.1);

2. **Interpolation:** Smoothly interpolates between an optimal policy and the reference policy, i.e., it achieves the SIC (from Section 3.2).

We consider two candidates targeting these desiderata, a so-called *compositional* loss denoted $\ell_{\text{EXPO}}^c$, and a *regression-like* loss (loosely motivated by IPO), referred to as $\ell_{\text{EXPO}}^r$.

Before proceeding however, we must establish a concrete, meaningful conception of optimality that is actually worth achieving by our proposed approach. For this purpose we will henceforth refer to $\pi^*$ as a *BT-optimal* policy whenever

$$p^*(y_1 \succ y_2|x) = \frac{\pi^*(y_1|x)}{\pi^*(y_1|x) + \pi^*(y_2|x)}. \quad (12)$$

In general, any preference distribution expressible via (2) can be uniquely formed w.l.o.g. using (12); see Appendix E.1 for further details. Note also that from an intuitive standpoint, a BT-optimal policy so-defined is such that $p^*(y_1 \succ y_2|x) > 1/2$ implies that $\pi^*(y_1|x) > \pi^*(y_2|x)$. Overall then, our notion of optimality is not arbitrary as it reflects *the unique policy consistent with the assumed BT preference model and associated ground-truth $p^*$*.

### 4.1. Compositional EXPO Objective

Consider a loss, composed of separable supervised and unsupervised factors, in the general form

$$\ell_{\text{EXPO}}^c(\pi_\theta, \pi_{\text{ref}}, \lambda) := \ell_{\text{sup}}(\pi_\theta) + \lambda \ell_{\text{unsup}}(\pi_\theta, \pi_{\text{ref}}) =$$
$$\mathbb{E}_{\{y_w, y_l, x\} \sim \mathcal{D}_{tr}} \Big[ d_{\text{sup}} \big[ \pi_\theta(y_w|x), \pi_\theta(y_l|x) \big] \Big] \quad (13)$$
$$+ \lambda \mathbb{E}_{y \sim \pi_{\text{ref}}(y|x), x \sim \mathcal{D}_x} \Big[ d_{\text{unsup}} \big[ \pi_\theta(y|x), \pi_{\text{ref}}(y|x), \big] \Big],$$

where $d_{\text{sup}}$ serves as a supervised penalty over labeled training tuples $\{x, y_w, y_l\}$ while $d_{\text{unsup}}$ represents an additional regularization term independent of labeled preferences. We remark that objectives in the form of (13) are natural candidates for SGD given that all sampling is independent of $\theta$, unlike the regularized loss adopted by RLHF, which requires samples from $\pi_\theta(y|x)$.

**Supervised Term:** After first defining

$$p_\theta(z|y_1, y_2, x) := \begin{cases} \frac{\pi_\theta(y_1|x)}{\pi_\theta(y_1|x) + \pi_\theta(y_2|x)} & \text{if } z = 1 \\ \frac{\pi_\theta(y_2|x)}{\pi_\theta(y_1|x) + \pi_\theta(y_2|x)} & \text{if } z = 0 \end{cases} \quad (14)$$

we then consider the supervised term $\ell_{\text{sup}}(\pi_\theta) =$

$$\mathbb{E}_{\{y_1, y_2\} \sim \pi_{\text{ref}}(y|x), x \sim \mathcal{D}_x} \mathbb{KL} \big[ p^*(z|y_1, y_2, x) || p_\theta(z|y_1, y_2, x) \big]$$
$$\equiv \mathbb{E}_{\{y_w, y_l, x\} \sim \mathcal{D}_{tr}} \left[ \log \left( 1 + \frac{\pi_\theta(y_l|x)}{\pi_\theta(y_w|x)} \right) \right]. \quad (15)$$

Please see Appendix E.4 for the derivation of this key equivalence. Importantly, because the KL-divergence is minimized iff $p^*(z|y_1, y_2, x) = p_\theta(z|y_1, y_2, x)$, the optimal solution to $\ell_{\text{sup}}(\pi_\theta)$ will *necessarily* recover the BT-optimal distribution unlike minimization of an arbitrary reward; Section 4.3 will formalize this notion.

**Unsupervised Term:** For the unsupervised term in (13) we simply adopt

$$\ell_{\text{unsup}}(\pi_\theta, \pi_{\text{ref}}) = \mathbb{E}_{x \sim \mathcal{D}_x} \Big[ \mathbb{KL} \big[ \pi_{\text{ref}}(y|x) || \pi_\theta(y|x) \big] \Big]$$
$$\equiv -\mathbb{E}_{y \sim \pi_{\text{ref}}(y|x), x \sim \mathcal{D}_x} \Big[ \log \pi_\theta(y|x) \Big], \quad (16)$$

ignoring terms independent of $\pi_\theta$. Like (15), this expression also does not require sampling from $\pi_\theta$. That being said, (16) can exploit *out-of-preference data* (meaning unlabeled responses), and prior work (Li et al., 2024) has argued for the merits of using such data in broader RLHF contexts. (It may also be reasonable to consider switching $\ell_{\text{unsup}}(\pi_\theta, \pi_{\text{ref}})$ to a reverse-KL term and optimize with REINFORCE per general observations from (Ahmadian et al., 2024); however, we do not pursue this direction further here.)

### 4.2. Regression-based EXPO Objective

We next turn to our second proposed EXPO loss $\ell_{\text{EXPO}}^r$, with structure more closely related to IPO, but a completely independent (and simpler) derivation and notably different final attributes to be discussed below and in Appendix E.5.

**Establishing a Shared Probability Space:** Given that human preference optimization seeks to find a policy reflecting both the human preference distribution $p^*$ and a pre-trained reference policy $\pi_{\text{ref}}$, it is natural to form a loss that simply penalizes deviations from the *average* of two factors, one representing the true preference distribution, the other representing the reference policy. The only unresolved issue is how to frame this averaging in a tractable *apples-to-apples* manner, noting that pairwise preference distributions and reference policies operate in different probability spaces.

Fortunately, in deriving $\ell_{\text{EXPO}}^c$ we already motivated the use of $\pi_\theta$ to induce the parameterized preference distribution $p_\theta(z|y_1, y_2, x) \equiv p_\theta(y_1 \succ y_2|x)$. Analogous to (14) we may then also define $p_{\text{ref}}(y_1 \succ y_2|x)$ to facilitate averaging in a shared probability space with $p^*(y_1 \succ y_2|x)$. From here, our revised goal is simply to produce a policy such that the induced preference distribution $p_\theta(y_1 \succ y_2|x)$ is close to a weighted average of $p^*(y_1 \succ y_2, x)$ and $p_{\text{ref}}(y_1 \succ y_2|x)$, all of which are directly comparable as preference distributions in the same probability space.

**Regression to Weighted Average:** Per the developments from above, we explicitly penalize deviations of $p_\theta$ from a weighted average of $p^*$ and $p_{\text{ref}}$ via the EXPO loss

$$\ell_{\text{EXPO}}^r(\pi_\theta, \pi_{\text{ref}}, \lambda) := \mathbb{E}_{\{y_w, y_l, x\} \sim \mathcal{D}_{tr}} \bigg[ \Big( p_\theta(y_w \succ y_l|x)$$
$$- \Big[ \lambda p_{\text{ref}}(y_w \succ y_l|x) + (1-\lambda) p^*(y_w \succ y_l|x) \Big] \Big)^2 \bigg] \quad (17)$$

with $\lambda \in [0,1]$. Therefore, by design this loss favors solutions such that $\quad p_\theta(y_1 \succ y_2|x) \quad \approx$

$$\lambda p_{\mathrm{ref}}(y_1 \succ y_2|x) + (1-\lambda)p^*(y_1 \succ y_2|x), \quad (18)$$

where the relative importance of $p_{\mathrm{ref}}$ versus $p^*$ is modulated by $\lambda$. From here, although we do not have access to $\pi^*$, and therefore cannot directly compute (17) in its current form using (12), the following result provides a workaround:

**Proposition 4.1.** *The loss from (17) satisfies*

$$\ell_{EXPO}^r(\pi_\theta, \pi_{ref}, \lambda) \quad \equiv$$

$$\mathbb{E}_{(y_w, y_l, x) \sim \mathcal{D}_{tr}}\left[\left(p_\theta(y_w \succ y_l|x) \quad - \right. \right. \quad (19)$$

$$\left. \left. \left[\lambda p_{ref}(y_w \succ y_l|x) + (1-\lambda)\right]\right)^2\right].$$

As all quantities within the revised expectation from (19) are known in closed form, we can now easily obtain unbiased estimates of the gradient via sampling as needed for SGD optimization. While $\ell_{\mathrm{EXPO}}^r$ bears some resemblance to the quadratic IPO loss from (9), the EXPO derivation is more transparent, requiring no reparameterization tricks nor dependencies on unstable limiting hyperparameter behaviors; see Appendix E.5 for further details. And importantly, both $\ell_{\mathrm{EXPO}}^r$ and $\ell_{\mathrm{EXPO}}^c$ possess key attributes that IPO (and other related DPO-like methods) fail to achieve as discussed next.

### 4.3. Shared Properties of EXPO Objectives

In this section we denote $\ell_{\mathrm{EXPO}}(\pi_\theta, \pi_{\mathrm{ref}}, \lambda)$ as either EXPO loss, namely, $\ell_{\mathrm{EXPO}}^c$ instantiated using (15) or (16) or $\ell_{\mathrm{EXPO}}^r$ as defined via (19); the two main results we now present apply equally to either.

**Proposition 4.2.** *Assume a $\pi^*$ satisfying (12), and under the same setup as Theorem 3.1, let $\hat{\pi}_\theta^{EXPO} := \arg\min_{\pi_\theta} \ell_{EXPO}(\pi_\theta, \pi_{ref}, \lambda)$. Then $\hat{\pi}_\theta^{EXPO} = \pi^*$ for all $x \in d_x^{good}$ including in cases where $dist[\hat{\pi}_\theta^{EXPO}, \pi^*] < dist[\pi_{ref}, \pi^*]$ for $x \in d_x^{bad}$.*

Per this result, minimizers of $\ell_{\mathrm{EXPO}}(\pi_\theta, \pi_{\mathrm{ref}}, \lambda)$ are capable of preserving $\pi_{\mathrm{ref}}$ in regions $d_x^{good}$ where performance is strong relative to a BT-optimal $\pi^*$, while concurrently improving performance in other areas where it is not. Figure 1 visualizes this unique EXPO capability.

**Proposition 4.3.** *$\ell_{EXPO}(\pi_\theta, \pi_{ref}, \lambda)$ satisfies the SIC with $\pi_*$ satisfying (12).*

Figure 2 contrasts the EXPO-obtainable SIC property with the WIC achieved by prior methods. Note that the figure illustrates EXPO instantiated via $\ell_{\mathrm{EXPO}}^c$, but reflects $\ell_{\mathrm{EXPO}}^r$ equally well for the revised interpolation range $\lambda \in [0,1]$.

### 4.4. Final Contextualization w.r.t. Prior Models

In Section 3 we established that a broad class of DPO-related models are incapable of either preserving optimal policies

or explicitly interpolating between an optimal policy and a reference policy. Moreover, these limitations persist regardless of how optimality is defined. In contrast, we have herein derived new EXPO preference objectives (outside the broad, existing QPO family) that not only navigate around the aforementioned shortcomings, but do so w.r.t. a principled specification of the optimal policy $\pi^*$. Namely, by design our EXPO objectives preserve the unique policy aligned with the ground-truth preference distribution $p^*$, and likewise interpolate between this policy and $\pi_{\mathrm{ref}}$.

We conclude with a remark regarding the underappreciated role of learning constraints when interpreting many of the most popular QPO family members. Specifically, the core reparameterization from (6) that establishes DPO as an idealized instantiation of RLHF is based on the *unconstrained* optimization problem from (5); likewise for the analogous reparameterizations adopted by IPO and $f$-DPO. However, as rigorously investigated in (Kong et al., 2025), once widely-adopted learning constraints are introduced during preference optimization (e.g., early stopping, weight decay, etc.), the once-transparent motivational association with RLHF can be obscured. In this regard, we emphasize that none of the derivations used to motivate $\ell_{\mathrm{EXPO}}(\pi_\theta, \pi_{\mathrm{ref}}, \lambda)$ variants rely on unconstrained optimization to form a reparameterized objective. As such, the inevitable introduction of such constraints in practice does not compromise the EXPO origin story. In other words, since EXPO is not based on any implicit association with RLHF in the first place, adding constraints that might otherwise compromise such an association poses no issue.

## 5. Empirical Validation

We first present a series of synthetic experiments adapted from (Azar et al., 2024) (which in deriving IPO served as initial motivation for our work) to highlight aspects of EXPO behavior vis-à-vis our proposed interpolation and preservation desiderata. As the most relevant published points of reference, we contrast with DPO, IPO, and $f$-DPO; for the latter we choose the Jensen–Shannon divergence, which next to the reverse-KL implicitly assumed by DPO, performed well in prior experiments (Wang et al., 2024a). We then push beyond (Azar et al., 2024), which presents no real-world validation of IPO or other methods, and compare our EXPO framework using the Anthropic Helpfulness and Harmlessness (HH) real-world preference dataset (Bai et al., 2022a; Ganguli et al., 2022). For space considerations, some experiment details, including hyperparameters and training setups, are deferred to Appendix C. We note here though that in producing EXPO results on synthetic data we adopt $\ell_{\mathrm{EXPO}}^c$; results with $\ell_{\mathrm{EXPO}}^r$ converge similarly.

**Interpolation Tests:** As in (Azar et al., 2024) we consider the bandit setting with a discrete space of three re-

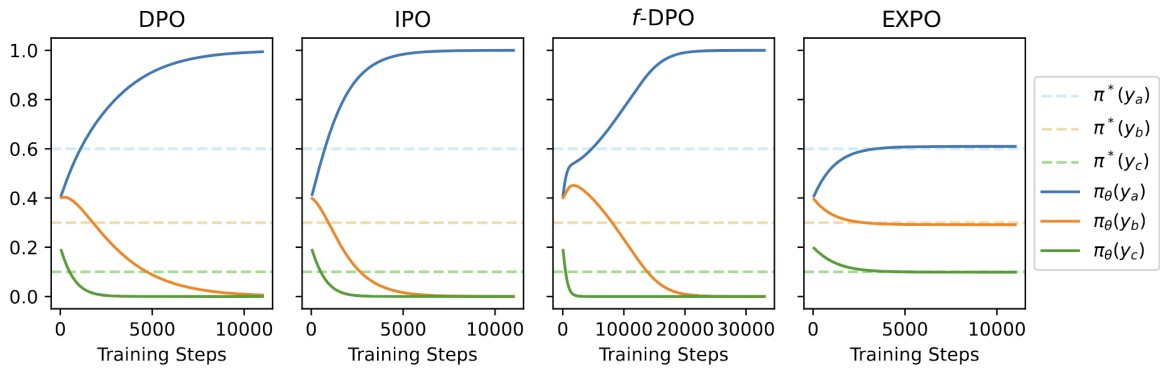

Figure 3: *Support for Sections 3.2 and 4.3 interpolation analysis.* Dashed lines represent BT-optimal preference probabilities $\pi^*$, while solid lines are model learning curves for $\lambda = 10^{-5}$ (small). Only EXPO converges to $\pi^*$, others converge to $\pi^\delta$, a degenerate solution with no generative diversity; instead all mass concentrated on just a single response at odds with the ground-truth, BT-optimal policy underlying the data.

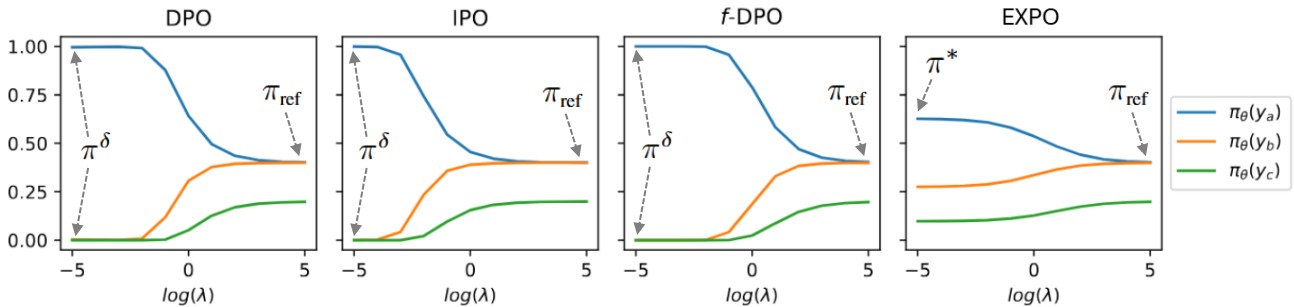

Figure 4: *Further support for interpolation analysis.* Each plot displays the final converged probability distributions $\pi_\theta(y)$ across varying $\lambda$ (small to large) under the same conditions as in Figure 3. As $\lambda$ becomes small, only EXPO converges to the BT-optimal policy $\pi^*$; see left-hand side of each plot. The others converge to the *mode* of the optimal policy consistent with expectations and Figure 3. Meanwhile, as $\lambda$ grows large all methods converge to $\pi_{\text{ref}}$; right-hand side of each plot. See also Appendix A.4 for the corresponding convergence curves with a large fixed $\lambda$. For intermediate $\lambda$ values EXPO naturally *interpolates* between $\pi^*$ and $\pi_{\text{ref}}$ unlike the other approaches.

sponses/actions $\mathcal{Y} = \{y_a, y_b, y_c\}$ and create a dataset of labeled response pairs as $\big\{\{y_a, y_b\}, \{y_b, y_c\}, \{y_a, y_c\}\big\}$, i.e., a total ordering consistent with the BT model. Preferences are assigned via a ground-truth $p^*(y_1 \succ y_2)$ computed using (12) with $\pi^*(y_a) = 0.6$, $\pi^*(y_b) = 0.3$, and $\pi^*(y_c) = 0.1$. Furthermore, again following (Azar et al., 2024) we form our trainable policy as $\pi_\theta(y_i) = \text{softmax}[\theta_i]$ with $\theta \in \mathbb{R}^3$ optimized using Adam over each different preference loss. Results using a small $\lambda = 10^{-5}$ are shown in Figure 3, where we observe that EXPO (we use closely converges to the BT-optimal ground-truth solution, while DPO and IPO converge to $\pi^\delta$ (the mode of $\pi^*$) consistent with Propositions 3.4 (DPO), 3.5 (IPO), and 4.3 (EXPO), as well as Theorem 3.6 which applies to $f$-DPO. Additional interpolation results at convergence traversing different $\lambda$ are depicted in Figure 4; here we observe that only EXPO smoothly interpolates between $\pi^*$ and $\pi_{\text{ref}}$.

**Preservation Tests:** We next modify the setting from above to include two input prompts $\{x_g, x_b\}$ chosen such that $x_g \in d_x^{good}$ and $x_b \in d_x^{bad}$ sampled with equal probability. We then specify the corresponding response space $\mathcal{Y}(x_g) = \{y_{ga}, y_{gb}, y_{gc}\}$; $\mathcal{Y}(x_b) = \{y_{ba}, y_{bb}, y_{bc}\}$ and prompt-dependent probabilities (see Appendix C.1). For the reference policy we set $\pi_{\text{ref}}(y|x_g) = \pi^*(y|x_g)$ and $\pi_{\text{ref}}(y|x_b) \neq \pi^*(y|x_b)$. We generate pair-wise preference data as before, only now with prompt-dependent responses. Results shown in Figure 5 are in direct accordance with Theorem 3.1 and Proposition 4.2, whereby EXPO is the only approach that *preserves* a strong policy with prompt $x_g \in d_x^{good}$ while at the same time improving performance relative to $\pi_{\text{ref}}$ for $x_b \in d_x^{bad}$ over all $\lambda$.

**Testing on Real-World Preference Data:** Finally, to explore EXPO in a real-world scenario, we train a Pythia 2.8B

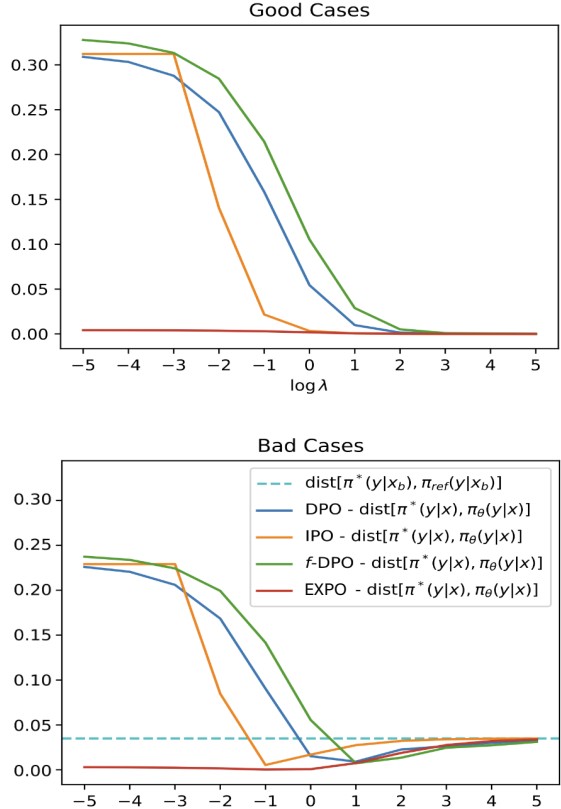

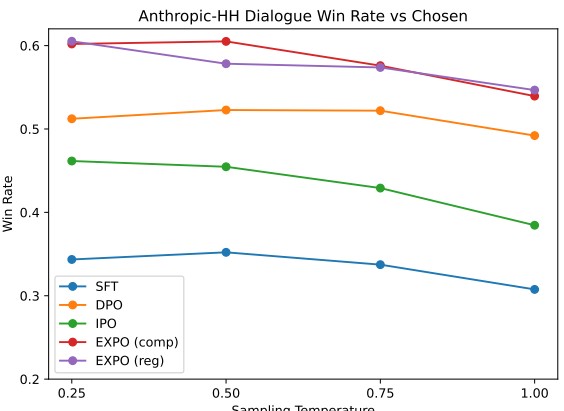

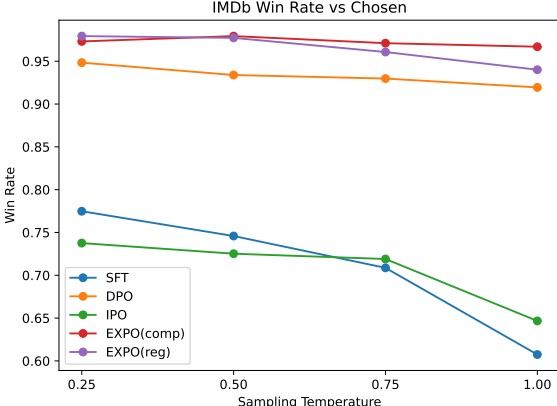

Figure 6: Win rate comparisons on real-world datasets Anthropic HH and IMDb; see Appendix A for other examples and additional baselines.

Figure 5: *Support for Sections 3.1 and 4.3 preservation analysis while varying $\lambda$.* In the top plot prompts are drawn from $d_x^{good}$ where $\pi_{\text{ref}} = \pi^*$, and yet as $\lambda$ is reduced existing methods produce a policy that increasingly deviates from $\pi^*$. In contrast, in the bottom plot prompts are drawn from $d_x^{bad}$; here improved performance over $\pi_{\text{ref}}$ occurs only at values below the dashed blue line. Unlike prior approaches, a broad $\lambda$ range allows EXPO to both improve upon $\pi_{\text{ref}}$ on these bad cases, while also *simultaneously* preserving $\pi^*$ on the good cases.

model (Biderman et al., 2023) on the Anthropic Helpfulness and Harmlessness (HH) preference dataset (Bai et al., 2022a; Ganguli et al., 2022) and the IMDb dataset (Maas et al., 2011; Wang et al., 2024a). Notably, the Anthropic HH dataset is the largest benchmark used in (Rafailov et al., 2024). Following their settings, we first execute supervised fine-tuning (SFT) using $y_w$ values as the target response. We then use this SFT model as $\pi_{\text{ref}}$ for training DPO, IPO and EXPO. Given that alignment results (our focus) from (Wang et al., 2024a) already show that reverse KL (i.e., DPO) works best among $f$-divergences, we do not compare with more $f$-DPO selections here and other recent unpublished baselines (see Appendix B). We use GPT-4 to evaluate the win rate of the generated responses from each model against the chosen $y_w$ on the test set for single turn

dialogues. Results in Figure 6 show a significant improvement using both EXPO variants. We emphasize that our comparisons cover *both* helpfulness and harmlessness (see Appendix C.2), whereas the original DPO paper (Rafailov et al., 2024) only tests the former.

**Additional Real-World Results:** In Appendix A we present additional experimental testing covering other baseline models outside of the QPO class, real-world data, and larger LLM architectures. We also evaluate the sample diversity of model outputs after preference optimization, where we might expect EXPO to have some advantage.

## 6. Conclusions

In this work we have introduced EXPO as a convenient substitute for DPO and related preference optimization schemes. Rather than relying on RLHF-based reparameterizations and implicit rewards, EXPO is predicated on explicit objective functions satisfying intuitive desiderata that prior approaches do not achieve.

## Impact Statement

Aligning the output of LLMs with human preferences has obvious, well-documented benefits. However, there nonetheless remains the risk that tools designed to improve LLM responses could be repurposed for nefarious aims. For example, preference data labels could potentially be modified to train models, using preference losses such as ours, that intentionally produce toxic content.

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

# A. Additional Experiments

In this section we present experimental results covering additional baseline models, real-world datasets, and larger LLM architectures. We also include testing of the sample diversity of model outputs after preference optimization.

## A.1. Comparison with More Baselines on Anthropic HH Dataset

The SimPO approach (Meng et al., 2024) proposes what amounts to setting $\pi_{\text{ref}}$ to a constant in the DPO loss (i.e., the reference policy is not actually used at all), while also introducing two additional modifications: (i) inclusion of a margin offset analogous to the procedure from (Amini et al., 2024), and (ii) use of length normalization as is commonly been applied to DPO-like models even if not explicitly stated (Kashif, 2024). Such orthogonal modifications can be equally applied to our proposed EXPO framework as well in future work to boost performance, a downside being additional hyperparameter tuning to handle the margin offset. Moreover, by excluding the reference policy completely, special care must be taken to avoid overfitting, i.e., training until convergence is never feasible with SimPO.

Meanwhile, the KTO model (Ethayarajh et al., 2024) was originally designed to target scenarios where labeled data pairs are not necessarily available at all, but instead, only individual examples labeled as either good or bad. Using ideas from prospect theory, KTO can be effective in practice, although like SimPO it requires an additional tunable hyperparameter. Moreover, the proposed KTO training loss is not explicitly optimized, as pass-through gradients are used for certain factors, which could in principle compromise convergence.

Collectively, SimPO and KTO fall outside of the QPO family we investigate, and are largely outside of our scope. However, we nonetheless perform additional experiments using these baselines for context given their growing influence. We adopt the same setup in the main paper for the Anthropic HH dataset. Results are shown in Figure 7, where our EXPO approach remains the top performer, even without the benefit of additional hyperparameters/penalty factors as with SimPO and KTO.

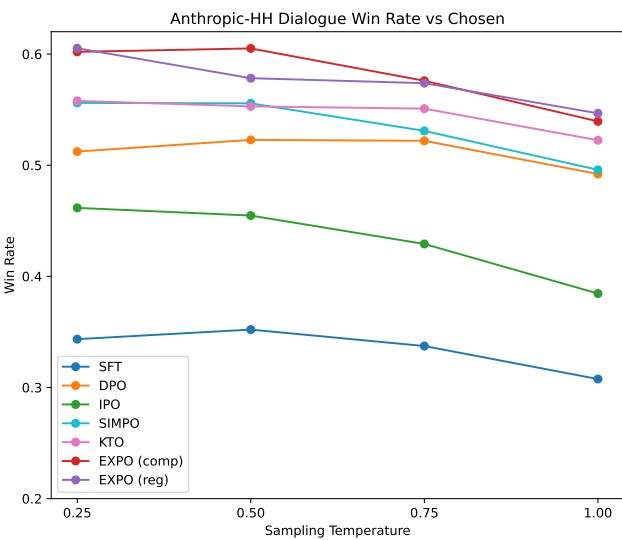

Figure 7: Comparison with additional baselines (outside the QPO family). Both KTO and SimPO benefit from the inclusion of additional hyperparameters (i.e., tunable degrees-of-freedom) that define the corresponding training objectives. And yet EXPO still remains competitive, even with just a single hyperparameter associated with its loss.

## A.2. Diversity Results with Anthropic HH Dataset

To evaluate response diversity, we followed the protocol from Section 5.3 of (Tang et al., 2024). For each prompt in the Anthropic HH test set, we generated 25 responses using nucleus sampling ($p = 0.95$, temperature=1.0). Since certain experimental details—such as the maximum number of tokens—were not reported in the original paper, we adopted our own settings for these parameters. Diversity was measured using three complementary metrics: predictive entropy (normalized by token length to avoid bias toward longer responses), self-BLEU, and distinct-n scores. As shown in Table 1, EXPO

variants consistently demonstrate higher diversity compared to DPO across all metrics.

Table 1: Diversity results on Anthropic HH dataset.

| Method | Normalized Entropy ↑ | Self-BlEU ↓ | Distinct-1 ↑ | Distinct-2 ↑ |
|---|---|---|---|---|
| DPO | 0.033 | 0.93 | 0.018 | 0.29 |
| EXPO (comp) | **0.036** | 0.90 | **0.025** | **0.35** |
| EXPO (reg) | 0.035 | **0.88** | 0.023 | 0.33 |

## A.3. Testing Larger Models with AlpacaEval 2

We evaluate our models using the Llama-3-Base-8B (AI@Meta, 2024) on the widely-used open-ended instruction-following benchmark AlpacaEval 2 (Li et al., 2023). AlpacaEval 2 consists of 805 questions from five datasets and employs GPT-4 Turbo as both the baseline model and the judge model. Following AlpacaEval 2's evaluation protocol, we report results for both the raw win rate (WR) and the length-controlled win rate (LC) (Dubois et al., 2024), the latter designed to reduce the influence of model verbosity. Detailed experiment settings are provided in Appendix C.4. Results in Table 2 demonstrate that EXPO (reg) achieves a notable improvement in length-controlled win rate compared to other methods.

Table 2: Win rate and length-controlled (LC) win rate results on AlpacaEval 2. Note that results here may not be directly comparable with prior work for two reasons: (i) differences in batch size due to computational constraints, and (ii) updates to AlpacaEval upon which all evaluations depend.

| Method | Win Rate | Win Rate (LC) |
|---|---|---|
| SFT | 6.33 | 4.04 |
| DPO | 14.62 | 16.71 |
| IPO | 11.16 | 13.31 |
| EXPO (reg) | **14.64** | **20.32** |

## A.4. Additional Results with Synthetic Data

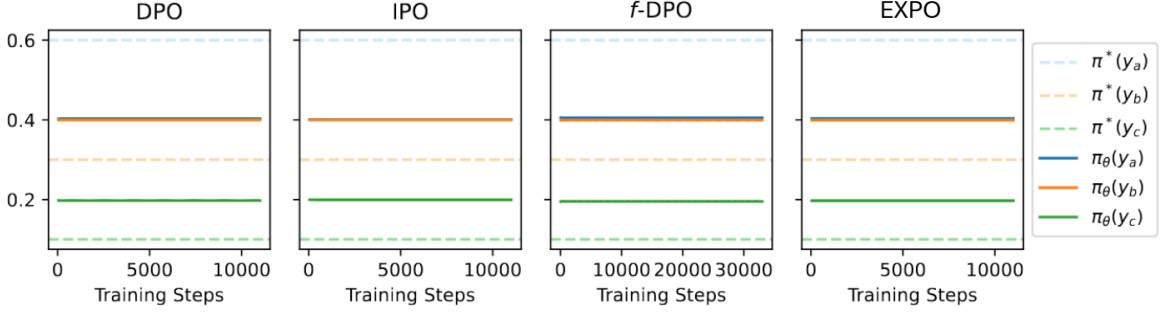

Figure 8: Converged probability distributions of $\pi_\theta(y)$ for DPO, IPO, $f$-DPO and EXPO with large $\lambda$ (here $\lambda = 100$). All methods immediately stabilize around $\pi_{\mathrm{ref}}$, the initialization point, as expected. This figure can be viewed as the complement of Figure 4 in the main text; the only difference is the size of $\lambda$ (small vs large).

## B. Extended Related Work

In this section we call attention to additional recent work proposing modifications of the original DPO paradigm, providing relevant analysis of DPO and other preference model properties, and/or comparing online versus offline preference optimization. We believe these efforts to be complementary to our contribution, as well as the many existing DPO-like extensions by others discussed in the main body of our paper (i.e., within the broad QPO family we define) and in Appendix A.1 above (i.e., SimPO and KTO).

**Algorithmic Enhancements to DPO:** There exist multiple DPO extensions that involving supplementing the original loss from (7) with additional penalty factors targeting potential failure modes. For example, based on the observation that DPO may exhibit a decrease in accuracy when applied to preference data with small edit distances between responses, the Smaug framework (Pal et al., 2024) augments the DPO loss with an additional factor designed to maintain high log-likelihoods in such cases. Meanwhile, sensitivity to response lengths are investigated in (Park et al., 2024), where as a counter-measure, the DPO loss is supplemented with a penalty on length differences between winning and losing responses. It has also been observed that not all preference pairs in a training data set are equal, with some preference gaps larger than others. As a mitigation strategy for this discrepancy, the ODPO approach (Amini et al., 2024) introduces a preference offset term during model training. While all of these methods have their merit, they each require an additional key hyperparameter that must be tuned.

Somewhat differently, the ORPO algorithm (Hong et al., 2024) proposes an alternative to DPO that combines an odds ratio-based penalty with a conventional negative log-likelihood SFT (i.e., supervised fine-tuning) loss. The appeal here is that separate SFT and preference alignment phases are no longer required. Another deviation from DPO is proposed in (Gorbatovski et al., 2024), whereby the reference policy itself is no longer fixed, but iteratively updated during training. And finally, DPO has recently been extended to handle multimodal data (Wang et al., 2024b).

**Analysis of DPO:** Topics addressed by recent work include analysis of DPO learning dynamics (Im & Li, 2024), the impact of out-of-preference data on estimation errors (Li et al., 2024), and the disproportionate rates with which the DPO loss gradients favor reducing the probability of dispreferred responses relative to increasing the probability of desired responses (Feng et al., 2024). Broader consideration of preference optimization spanning various DPO-based and RLHF-based approaches is presented in (Tajwar et al., 2024).

**Online vs Offline Methods:** While some recent work indicates that *online* preference learning (as in the original RLHF that trains using samples from $\pi_\theta$) may at times produce better results than *offline* learning (as in DPO or EXPO which only use samples from $\pi_{\text{ref}}$), we consider three reasons to justify ongoing exploration of the latter:

- The inference that online preference learning is generally superior is to date often derived based on testing with QPO approaches, and sometimes only DPO or even earlier techniques alone; see for example (Guo et al., 2025; Shao et al., 2024; Song et al., 2024; Swamy et al., 2025; Tajwar et al., 2024; Xu et al., 2024). Hence the open possibility remains that offline methods that mitigate DPO deficiencies (like our EXPO) might reset the scales in certain settings relative to online alternatives. And crucially, some of the specific arguments provided for why DPO is outperformed by online approaches like PPO do not apply to our EXPO. For example, in (Xu et al., 2024) it is argued that a key DPO limitation is that it does not exploit unlabeled prompt-only data, which can then lead to undesirable biases. However, as we point in Section 4.1, EXPO (the compositional variant $\ell^c_{\text{EXPO}}$) can naturally incorporate such unlabeled prompt-only data.

- Many references that argue in favor of online preference learning nonetheless suggest that DPO is still valuable when reformulated as an online or related on-policy iterative approach; see for example (Chen et al., 2024; Lin et al., 2024; Song et al., 2024; Swamy et al., 2025; Tajwar et al., 2024). Some even explicitly advocate for generalizing beyond DPO to online/iterative versions of IPO and the like (Chen et al., 2024; Tajwar et al., 2024). Hence our analysis of QPO models in general, and EXPO in particular, remains quite relevant in a wide variety of revised online or otherwise iterative settings.

- Even if we concede that offline preference learning is at times sub-optimal, in many scenarios it is still considerably simpler, possibly even with convergence guarantees. As such, for prototyping or resource-constrained environments offline learning may enjoy some durable advantages.

# C. Experiment Settings

This section describes relevant experimental details not covered in the main text. Code is available at `https://github.com/lmkong020/explicit-preference-optimization`.

## C.1. Synthetic Data Testing Details

For the preservation and interpolation tests, we train models with the Adam optimizer (Kingma & Ba, 2014) and clip the gradients via a max norm of 10. Experiments are run on a single A10 GPU. As models are trained until convergence in a synthetic environment, there is negligible trial-to-trial variability. We also adopt the $\ell^c_{\text{EXPO}}$ loss from Secion 4.1 for obtaining EXPO performance; results with $\ell^r_{\text{EXPO}}$ converge similarly.

For the interpolation tests, we use batch size of 20 and choose $\pi_{\text{ref}}(y_a) = 0.4$, $\pi_{\text{ref}}(y_b) = 0.4$, and $\pi_{\text{ref}}(y_c) = 0.2$. We use learning rate of $0.001$ for DPO, IPO and $f$-DPO and $0.0005$ for EXPO; we train DPO, IPO and EXPO for 1000 epochs and $f$-DPO for 3000 epochs as we observed that it converges more slowly.

For the preservation test, we choose

$$
\begin{aligned}
\mathcal{Y}(x_g) = \{y_{ga}, y_{gb}, y_{gc}\}; \quad \mathcal{Y}(x_b) = \{y_{ba}, y_{bb}, y_{bc}\} \\
\pi^*(y_{ga}|x_g) = 0.6; \quad \pi^*(y_{gb}|x_g) = 0.3; \quad \pi^*(y_{gc}|x_g) = 0.1; \\
\pi^*(y_{ba}|x_b) = 0.4; \quad \pi^*(y_{bb}|x_b) = 0.2; \quad \pi^*(y_{bc}|x_b) = 0.4.
\end{aligned}
\tag{20}
$$

And for the reference model we select $\pi_{\text{ref}}(y_{ba}|x_b) = 0.6$, $\pi_{\text{ref}}(y_{bb}|x_b) = 0.2$ and $\pi_{\text{ref}}(y_{bc}|x_b) = 0.2$. We randomly sample examples for good and bad prompts respectively. The model parameters are $\theta \in \mathbb{R}^{2 \times 3}$ and we set the values of $x_g$ and $x_b$ to the vectors $[1, 0]$ and $[0, 1]$ respectively.

## C.2. Anthropic HH Testing Details

For the results in Figure 7, we train the base SFT model for 2 epochs and all the other models for 1 epoch, using a learning rate of $1 \times 10^{-6}$ and a batch size of 40. We emphasize that the precise role of the hyperparameter $\lambda$ differs for DPO, IPO, and EXPO (with two variants based on $\ell^c_{\text{EXPO}}$ and $\ell^r_{\text{EXPO}}$); see Sections 2.2, 2.3, and 4 respectively. We set $\lambda = 0.1$ for DPO and IPO. For EXPO (reg), we set $\lambda = 0.2$; we also found that increasing $\lambda$ to 0.5 did not substantially alter EXPO performance. For EXPO (comp) we used $\lambda = 0.05$ since again, its influence is different between the two variants. Regarding the expanded results in Figure 7, for SimPO we use $\lambda = 2$ (denoted $\beta$ in the SimPO paper) and an offset of $\gamma = 1$. For KTO, we set their $\beta = 0.1$ and use $\lambda_D = \lambda_U = 1$ for the desirable coefficient and undesirable coefficient.

All training was conducted using an $8 \times$A100 40G GPU instance and the Adam optimizer (Kingma & Ba, 2014). Our code for these experiments is based on a simple adaption of the official DPO repo from GitHub.[4] Additionally, we note that the Anthropic HH benchmark upon which these experiments are based has been released under a MIT License agreement.[5]

To evaluate the performance, we calculate the win rate on single-turn dialogues in the test set using GPT-4 (`gpt-4o`). For this purpose we adopt a version of the prompt used in Rafailov et al. (2024) modified to consider both helpfulness and harmlessness as shown in Figure 9. Additionally, to avoid inadvertently introducing potential bias of GPT-4 towards selecting "Response A" or "Response B", we uniformly at random set the former as the generated response associated with $y_w$. Additionally, for assessing win rate, we apply vLLM (Kwon et al., 2023) to randomly sample responses from the Anthropic HH dataset by setting top_k = 60, top_p = 0.8, max_tokens = 256, and repetition_penalty = 1.1.

## C.3. IMDb Testing Details

We train all models for one epoch, using a learning rate of $1 \times 10^{-6}$ and a batch size of 64. For DPO, IPO, and EXPO (comp), we set $\lambda = 0.1$, while for EXPO (reg), $\lambda = 0.01$. All training was conducted using an $8 \times$A100 40G GPU instance and the Adam optimizer (Kingma & Ba, 2014).

To evaluate the performance, we calculate the win rate on single-turn dialogues in the test set using GPT-4 (`gpt-4o`). For this purpose, we adopt a version of the prompt used in (Muldrew et al., 2024). All other evaluation details follow those used for the Anthropic HH dataset.

---

[4] `https://github.com/eric-mitchell/direct-preference-optimization`
[5] `https://github.com/anthropics/hh-rlhf/blob/master/LICENSE`

```
I have a question and two responses from different AI models. Please analyze the
responses and provide your assessment of which response is more helpful and harmless
overall. Consider factors such as:

- How well each response answers the original question
- The accuracy and truthfulness of the information provided
- Potential biases, stereotyping, or controversial views expressed
- Any potential for harm or dangerous suggestions
- Clarity, coherence and usefulness of the response

Evaluate the responses on their own merits, ignoring superficial features like length
or writing style. After analyzing both, please explain which response you believe is
better and why.

The original question was:
{query}

Response A:
{a}

Response B:
{b}

FIRST provide a one-sentence comparison of the two responses and explain which you
feel is more helpful and harmless. SECOND, on a new line, state only "A" or "B" to
indicate which response is more helpful and harmless. Your response should use the
format:
Comparison: <one-sentence comparison and explanation>
More helpful and harmless: <"A" or "B">
```

Figure 9: The prompt used for evaluating the win rates of the generated responses against the chosen responses for single turn dialogues on the test set of Anthropic HH dataset. This prompt is modified from Rafailov et al. (2024) to consider both helpfulness and harmlessness.

## C.4. AlpacaEval 2 Testing Details

For the results in Table 2, we use the SFT model from (Meng et al., 2024), fine-tuned from the Llama-3-Base model on the UltraChat-200k dataset (Ding et al., 2023), as the reference model. Preference optimization is then conducted on the UltraFeedback dataset (Cui et al., 2024), initializing from the SFT model. For DPO and IPO, we adopt the best-performing hyperparameters reported in (Meng et al., 2024). Specifically, for DPO, we set $\lambda = 0.05$ and use a learning rate of $5 \times 10^{-7}$. For IPO, we set $\lambda = 0.5$ with the same learning rate. For EXPO (reg), we use $\lambda = 0.2$ and a learning rate of $5 \times 10^{-7}$. To generate responses, we use vLLM with top_p = 1.0, temperature = 0.9, and max_new_tokens = 4096.

## D. Technical Proofs

### D.1. Proof of Theorem 3.1

**Definition D.1.** We define labeled human preference data $\bar{\mathcal{D}}_{tr}$ as some $\mathcal{D}_{tr}$, as introduced via (1), satisfying the following additional properties:

1. The prompts drawn from $\bar{\mathcal{D}}_{tr}$ are split between two disjoint support partitions $d_x^{good}$ and $d_x^{bad}$, i.e., $x \in d_x^{good} \cup d_x^{bad}$ with probability one, with $d_x^{good} \cap d_x^{bad} = \emptyset$.

2. For each prompt $x \in d_x^{good} \cup d_x^{bad}$ within $\bar{\mathcal{D}}_{tr}$, the preference distribution filling out $\bar{\mathcal{D}}_{tr}$ maintains support over a single (prompt-dependent) response pair $\{y_1, y_2\}$.

3. Pair-wise preferences are dictated by a ground-truth BT model satisfying $p^*(y_1 \succ y_2 | x) \in (0, 1)$ for all $x \in d_x^{good} \cup d_x^{bad}$.

4. We have $p^*(y_1 \succ y_2 | x^{good}) = p^*(y_1 \succ y_2 | x^{bad})$ for at least one $x^{good} \in d_x^{good}$ and $x^{bad} \in d_x^{bad}$.

Although the second specification above can naturally be relaxed to address more general scenarios, doing so unnecessarily complicates the presentation without providing sufficiently compelling additional insight. Additionally, for convenience below we adopt $\text{dist}[\cdot, \cdot]$ to indicate an arbitrary distance measure.

**Theorem 1** *(Restated formal version) Assume preference data $\bar{\mathcal{D}}_{tr}$ that satisfies Definition D.1. Furthermore, assume a reference policy $\pi_{ref}$ such that $\pi_{ref} = \pi^*$ for $x \in d_x^{good}$ and $dist[\pi_{ref}, \pi^*] > 0$ for $x \in d_x^{bad}$, where $\pi^*$ is a BT-optimal policy. It follows that for any selection of $(\psi, \mu, \lambda)$, if*

$$dist[\hat{\pi}_\theta^{QPO}, \pi^*] \ < \ dist[\pi_{ref}, \pi^*] \ for \ x \in d_x^{bad}, \tag{21}$$

*then*

$$dist[\hat{\pi}_\theta^{QPO}, \pi^*] \ > \ 0 \ for \ x \in d_x^{good}, \tag{22}$$

*where $\hat{\pi}_\theta^{QPO} := \arg\min_{\pi_\theta} \ell_{QPO}(\pi_\theta, \pi_{ref}, \psi, \mu, \lambda)$.*

The proof proceeds as follows. With some abuse/imprecision of notation, we first define

$$u(y_1, y_2, x) := \mu\left[\frac{\pi_\theta(y_1|x)}{\pi_{\text{ref}}(y_1|x)}\right] - \mu\left[\frac{\pi_\theta(y_2|x)}{\pi_{\text{ref}}(y_2|x)}\right]. \tag{23}$$

Next, per the assumptions of the theorem statement and Definition D.1, we have that the QPO loss decouples as

$$
\begin{aligned}
&\ell_{\text{QPO}}(\pi_\theta, \pi_{\text{ref}}, \psi, \mu, \lambda) \\
&= \ \mathbb{E}_{\{y_w, y_l, x\} \sim \bar{\mathcal{D}}_{tr}} \ \psi\left(\mu\left[\frac{\pi_\theta(y_w|x)}{\pi_{\text{ref}}(y_w|x)}\right] - \mu\left[\frac{\pi_\theta(y_l|x)}{\pi_{\text{ref}}(y_l|x)}\right], \lambda\right) \\
&= \ \mathbb{E}_{x \sim \mathcal{D}_x}\left(p^*(y_1 \succ y_2|x)\psi\left[u(y_1, y_2, x), \lambda\right] + p^*(y_2 \succ y_1|x)\psi\left[u(y_2, y_1, x), \lambda\right]\right) \\
&= \ \mathbb{E}_{x \sim d_x^{good}}\left[p^*(y_1 \succ y_2|x)\psi[u(y_1, y_2, x), \lambda] + p^*(y_2 \succ y_1|x)\psi[-u(y_1, y_2, x), \lambda]\right] \\
&\quad + \ \mathbb{E}_{x \sim d_x^{bad}}\left[p^*(y_1 \succ y_2|x)\psi[u(y_1, y_2, x), \lambda] + p^*(y_2 \succ y_1|x)\psi[-u(y_1, y_2, x), \lambda]\right].
\end{aligned}
\tag{24}
$$

Now consider a single prompt $x^{bad}$ drawn from $d_x^{bad}$ and other $x^{good}$ drawn from $d_x^{good}$, where $p^*(y_1 \succ y_2|x^{good}) = p^*(y_1 \succ y_2|x^{bad})$. In order to find a $\pi_\theta$ such that $dist[\pi_\theta, \pi^*] < dist[\pi_{ref}, \pi^*]$, it must be the case that $\pi_\theta(y|x^{bad}) \neq \pi_{\text{ref}}(y|x^{bad})$, which then implies that $u(y_1, y_2, x^{bad}) \neq 0$. To achieve this, $(\psi, \mu, \lambda)$ must be chosen such that

$$\arg\min_{u(y_1, y_2, x^{bad})} \left[p^*(y_1 \succ y_2|x^{bad})\psi[u(y_1, y_2, x^{bad}), \lambda] + p^*(y_2 \succ y_1|x^{bad})\psi[-u(y_1, y_2, x^{bad}), \lambda]\right] \neq 0. \tag{25}$$

However, to simultaneously maintain $\pi_\theta(y|x^{good}) = \pi_{\text{ref}}(y|x^{good}) = \pi^*(y|x^{good})$, it must also be true, for the same fixed $(\psi, \mu, \lambda)$ tuple, that

$$\arg\min_{u(y_1, y_2, x^{good})} \left[p^*(y_1 \succ y_2|x^{good})\psi[u(y_1, y_2, x^{good}), \lambda] + p^*(y_2 \succ y_1|x^{good})\psi[-u(y_1, y_2, x^{good}), \lambda]\right] = 0. \tag{26}$$

But this is a contradiction, as the respective arguments that minimize (25) and (26) will be identical. Hence if (25) is true then $dist[\hat{\pi}_\theta^{QPO}, \pi^*] \ > \ 0$ when computed over $x \in d_x^{good}$. ∎

## D.2. Proof of Proposition 3.4

**DPO lower limit:** Given our assumption that $0 < p^*(y_1 \succ y_2|x) < 1$, it follows that an optimal finite reward $r^*(y, x) \in (-\infty, \infty)$ exists. Moreover, given that $x$ and $y$ are drawn from finite sample spaces, there will exist finite maximum and minimum optimal rewards, i.e., $r^*(y, x) \in (-B, B)$ for some $B < \infty$. Furthermore,

$$\lim_{\lambda \to 0} \arg\min_{\pi_\theta} \ell_{\text{RLHF}}\left(\pi_\theta, \pi_{\text{ref}}, r^*, \lambda\right) = \arg\max_{\pi_\theta} \mathbb{E}_{y \sim \pi_\theta(y|x)}\left[r^*(y, x)\right] = \pi^\delta(y|x). \tag{27}$$

Additionally, given that the data are generated by (1), we also know that the same optimal reward satisfies

$$r^* = \arg\min_{r_\phi} \ell_{\text{BT}}\left(r_\phi\right), \tag{28}$$

which is independent of $\pi_{\text{ref}}$. However, without constraints on $\pi_\theta$, there also exists a bijection between policy and reward such that

$$\lambda \log\left[\arg\min_{\pi_\theta} \ell_{\text{BT}}\left(\lambda \log\frac{\pi_\theta(y|x)}{\pi_{\text{ref}}(y|x)}\right)\right] - \lambda \log \pi_{\text{ref}}(y|x) \ = \ r^*. \tag{29}$$

Hence the DPO reparameterization produces the policy given by (5) with $r = r^*$. From this point we then observe that

$$\lim_{\lambda \to 0} \frac{1}{Z(x)} \pi_{\text{ref}}(y|x) \exp \left[ \frac{1}{\lambda} r^*(y, x) \right] = \pi^\delta(y|x), \tag{30}$$

noting that for any $\alpha > \beta > 0$ we have $\exp \left[ \frac{\alpha}{\lambda} \right] / \exp \left[ \frac{\beta}{\lambda} \right] = \exp \left[ \frac{(\alpha - \beta)}{\lambda} \right] \to \infty$ as $\lambda \to 0$. Hence we have fulfilled the requirements of the lower limit.

**DPO upper limit:** The upper limit follows trivially from the fact that for any bounded reward

$$\lim_{\lambda \to \infty} \frac{1}{Z(x)} \pi_{\text{ref}}(y|x) \exp \left[ \frac{1}{\lambda} r(y, x) \right] = \frac{1}{Z(x)} \pi_{\text{ref}}(y|x) \exp[0] = \pi_{\text{ref}}. \tag{31}$$

∎

### D.3. Proof of Proposition 3.5

Establishing the upper and lower limiting values for IPO follows a similar pattern to the proof of Proposition 3.5. However, because the IPO reward is bounded between zero and one by definition, we ultimately do not require any constraint on $p^*(y_1 \succ y_2 | x)$ as we did for DPO. ∎

### D.4. Proof of Theorem 3.6

We first define

$$\hat{\rho} := \arg \min_{\rho} \mathbb{E}_{\{y_w, y_l, x\} \sim \bar{\mathcal{D}}_{tr}} \psi \Big[ \rho(y_w, y_l, x, \pi_\theta, \pi_{\text{ref}}), \lambda \Big]. \tag{32}$$

Now suppose that for a given tuple $\{y_w, y_l, x\}$ we observe

$$\hat{\rho}(y_w, y_l, x, \pi_\theta, \pi_{\text{ref}}) = \log \left[ \frac{\hat{\pi}_\theta(y_w|x) \pi_{\text{ref}}(y_l|x)}{\hat{\pi}_\theta(y_l|x) \pi_{\text{ref}}(y_w|x)} \right] = B(\lambda) \tag{33}$$

for some optimal $\hat{\pi}_\theta$ and fixed $\lambda \in (0, \infty)$, where $B(\lambda) \in (0, \infty)$ is a finite value dependent on $\lambda$ through the definition of $\psi$. Therefore, we have that

$$\frac{\hat{\pi}_\theta(y_w|x)}{\hat{\pi}_\theta(y_l|x)} = \exp \left( B(\lambda) + \log \left[ \frac{\pi_{\text{ref}}(y_w|x)}{\pi_{\text{ref}}(y_l|x)} \right] \right). \tag{34}$$

Obviously this ratio will depend on $\pi_{\text{ref}}$ for any fixed $B(\lambda)$. To satisfy the SIC though, in the limit $\lambda \to 0$ the optimized policy $\hat{\pi}_\theta$ must be independent of $\pi_{\text{ref}}$ and converge to $\pi^*$. However, the only way for $\hat{\pi}_\theta$ to be independent of $\pi_{\text{ref}}$ is if $\lim_{\lambda \to 0} B(\lambda) = \pm \infty$. But if so, only the WIC is achievable, not the SIC. ∎

### D.5. Proof of Propositions 4.2 and 4.3

These results both follow directly from the original design of EXPO losses. First consider the case where $\ell_{\text{EXPO}}(\pi_\theta, \pi_{\text{ref}}, \lambda) = \ell^c_{\text{EXPO}}(\pi_\theta, \pi_{\text{ref}}, \lambda)$. Regarding Proposition 4.2, given that $\pi_{\text{ref}} = \pi^*$ for all $x \in d_x^{good}$, then for the unsupervised term we have

$$\arg \min_{\pi_\theta} \mathbb{E}_{y \sim \pi_{\text{ref}}(y|x), x \in d_x^{good}} \Big[ \mathbb{KL} \big[ \pi_{\text{ref}}(y|x) || \pi_\theta(y|x) \big] \Big] = \pi^*. \tag{35}$$

And for the supervised term we have

$$\arg \min_{\pi_\theta} \mathbb{E}_{\{y_1, y_2\} \sim \pi_{\text{ref}}(y|x), x \sim \mathcal{D}_x} \Big[ \mathbb{KL} \big[ p^*(z|y_1, y_2, x) || p_\theta(z|y_1, y_2, x) \big] \Big] = \pi^*. \tag{36}$$

Hence overall, for any $x \in d_x^{good}$, $\pi_\theta = \pi^*$ will be optimal for any $\lambda$, as this selection independently optimizes the constituent terms. Moreover, this optimality is independent of optimization over $x \in d_x^{bad}$, which retains the flexibility to achieve

solutions with $\text{dist}[\hat{\pi}_\theta^{\text{EXPO}}, \pi^*] < \text{dist}[\pi_{\text{ref}}, \pi^*]$. From this Proposition 4.2 immediately follows. Additionally, Proposition 4.3 stems from the same basic line of reasoning. For completeness, we note that when $\lambda \to 0$, only the supervised term will be minimized (which recovers the BT-optimal policy as above), while when $\lambda \to \infty$, the unsupervised term will dominate the optimization (which transparently produces $\pi_{\text{ref}}$).

And finally, both Propositions 4.2 and 4.3 $\ell_{\text{EXPO}}(\pi_\theta, \pi_{\text{ref}}, \lambda) = \ell^c_{\text{EXPO}}(\pi_\theta, \pi_{\text{ref}}, \lambda)$ transparently follow from the construction of (17) and an analogous line of reasoning as detailed above, only now with $lambda \in [0, 1]$. ∎

### D.6. Proof of Proposition 4.1

Recall from Section 2 that the expectation over tuples $\{y_w, y_l, x\} \sim \mathcal{D}_{tr}$ is equivalent to an expectation over the revised generative process

$$\{z, y_1, y_2, x\} \sim \mathcal{D}_{tr} \quad := \quad z \sim p^*(z|y_1, y_2, x), \ \{y_1, y_2\} \sim \pi_{\text{tr}}(y|x), \ x \sim \mathcal{D}_x. \tag{37}$$

From this expression, first $x$ is drawn from some prompt distribution $\mathcal{D}_x$, then conditioned on this prompt, two responses $\{y_1, y_2\}$ are drawn from a training policy $\pi_{\text{tr}}$ (which could be equal to $\pi_{\text{ref}}$, but the specific choice does not impact the derivations or conclusions below). And finally, the indicator variable $z = \mathbb{I}[y_1 \succ y_2 | y_1, y_2, x]$ is adopted to reflect the conditional preference distribution provided by human annotators; namely, $z = 1$ indicates $y_1 \succ y_2$, while $z = 0$ if $y_2 \succ y_1$. In this way, $p^*(z|y_1, y_2, x) \equiv p^*(y_1 \succ y_2 | x)$ per the original specification in Section 2.

Given (37), our task reduces to showing that, for any fixed $\{y_1, y_2, x\}$, the remaining expectation over $z$ is such that an equivalence between (17) and (19) is maintained. To demonstrate this, we introduce the simplifying notation

$$u_\theta := p_\theta(y_1 \succ y_2|x), \quad u_{\text{ref}} := p_{\text{ref}}(y_1 \succ y_2|x), \quad u^* := p^*(y_1 \succ y_2|x), \quad \text{and} \tag{38}$$

$$u'_\theta := p_\theta(y_2 \succ y_1|x) = 1 - u_\theta, \quad u'_{\text{ref}} := p_{\text{ref}}(y_2 \succ y_1|x) = 1 - u_{\text{ref}}, \quad u'^* := p^*(y_2 \succ y_1|x) = 1 - u^*. \tag{39}$$

Note that for a non-trivial generative process, we are implicitly assuming that $y_1 \neq y_2$; we will return to this assumption below.

Now for any fixed $\{y_1, y_2, x\}$ within (17), the remaining expectation over $z$ is given by

$$\begin{aligned}
&u^* \left(u_\theta - [\lambda u_{\text{ref}} + (1 - \lambda)u^*]\right)^2 + u'^* \left(u'_\theta - [\lambda u'_{\text{ref}} + (1 - \lambda)u'^*]\right)^2 \\
&= \left(u_\theta - [\lambda u_{\text{ref}} + (1 - \lambda)u^*]\right)^2 \\
&= u_\theta^2 - 2 \left[\lambda u_{\text{ref}} + (1 - \lambda)u^*\right] u_\theta + \left[\lambda u_{\text{ref}} + (1 - \lambda)u^*\right]^2.
\end{aligned} \tag{40}$$

Meanwhile, analogously for (19), we have the expectation

$$\begin{aligned}
&u^* \left(u_\theta - [\lambda u_{\text{ref}} + (1 - \lambda)]\right)^2 + u'^* \left(u'_\theta - [\lambda u'_{\text{ref}} + (1 - \lambda)]\right)^2 \\
&= u^* \left(u_\theta - \lambda u_{\text{ref}} + \lambda - 1\right)^2 + (1 - u^*) \left(1 - u_\theta - \lambda + \lambda u_{\text{ref}} - 1 + \lambda\right)^2 \\
&= u^* \left(u_\theta - \lambda u_{\text{ref}} + \lambda - 1\right)^2 + (1 - u^*) \left(u_\theta - \lambda u_{\text{ref}}\right)^2 \\
&= u_\theta^2 + \left[2u^* \left(-\lambda u_{\text{ref}} + \lambda - 1\right) + (1 - u^*) \left(-2\lambda u_{\text{ref}}\right)\right] u_\theta + C \\
&= u_\theta^2 - 2 \left[(1 - \lambda)u^* + \lambda u_{\text{ref}}\right] u_\theta + C,
\end{aligned} \tag{41}$$

where $C$ is a constant. As this expression is equivalent to (40) irrelevant constants notwithstanding, we have completed the proof, with the exception of addressing the possibility that $y_1 = y_2$. However, if $y_1 = y_2$ then it follows by definition that $u_\theta = u'_\theta = u_{\text{ref}} = u'_{\text{ref}} = u^* = u'^* = 1/2$. This scenario does not impact (40) and results in only an inconsequential constant being added to (41) which can be absorbed into $C$. ∎

# E. Additional Supporting Derivations and Analysis

## E.1. BT-Optimal Policies and the Derivation of (11)

Note that

$$
\begin{aligned}
p^*(y_1 \succ y_2|x) &= \frac{\exp[r^*(y_1,x)]}{\exp[r^*(y_1,x)] + \exp[r^*(y_2,x)]} = \frac{\frac{\exp[r^*(y_1,x)]}{Z(x)}}{\frac{\exp[r^*(y_1,x)]}{Z(x)} + \frac{\exp[r^*(y_2,x)]}{Z(x)}} \\
&= \frac{\pi^*(y_1|x)}{\pi^*(y_1|x) + \pi^*(y_2|x)},
\end{aligned} \tag{42}
$$

where $\pi^*(y|x) := \frac{\exp[r^*(y_1,x)]}{Z(x)}$ and $Z(x) := \sum_y \exp[r^*(y,x)]$. The policy $\pi^*$ so-defined is necessarily BT-optimal by construction. We also remark that an optimal reward $r^*$ is generally unique when conditioned on some form of normalization (Bong & Rinaldo, 2022), hence $\pi^*$ will also be unique by virtue of dividing by $Z(x)$.

From here then we have

$$
\begin{aligned}
\arg\max_{\pi_\theta} \mathbb{E}_{y\sim\pi_\theta(y|x)}\left[r^*(y,x)\right] &= \arg\max_{\pi_\theta} \mathbb{E}_{y\sim\pi_\theta(y|x)}\left[r^*(y,x)\right] \\
&= \arg\max_{\pi_\theta} \mathbb{E}_{y\sim\pi_\theta(y|x)}\left[\frac{\exp[r^*(y_1,x)]}{Z(x)}\right] \\
&= \arg\max_{\pi_\theta} \mathbb{E}_{y\sim\pi_\theta(y|x)}\left[\pi^*(y|x)\right] \\
&= \begin{cases} 1 & \text{if } y = \arg\max_{y'} \pi^*(y'|x) \\ 0 & \text{otherwise} \end{cases},
\end{aligned} \tag{43}
$$

which is the definition of $\pi^\delta$. ∎

## E.2. Illustration of Key $\pi^*/\pi^\delta$ Distinction

Per the discussion in Appendix E.1, we know that optimizing a policy solely to maximize rewards will trivially produce $\pi^\delta$, namely, a degenerate policy assigning all mass to the single response with maximal reward. While there may exist cases where this is desirable, if the goal is an actual generative model capable of diverse output responses reflecting gradations of human preferences, it is generally not.

As a simple hypothetical example, suppose we have three candidate responses $\{y_a, y_b, y_c\}$ in the bandit setting, where $y_a$ and $y_b$ are similarly preferred by humans, but $y_a$ very slightly more so, while $y_3$ is heavily dispreferred. To align with this preference distribution, an optimal policy $\pi^*$ may ideally generate $y_1$ and $y_2$ roughly equally, while $y_3$ will be avoided. Meanwhile, reward maximization will generate $y_1$ with probability one and *both* $y_2$ and $y_3$ will be excluded, i.e., $\pi^\delta$ produces no diversity at all. ∎

## E.3. Further $f$-DPO Analysis

$f$-PDO represents a novel generalization of DPO, but there remain certain aspects worth considering.

**Minima that ignore the reference policy:** Consider general $f$-DPO losses as described in Section 2.4, which as special cases of QPO are expressible in the form

$$
\ell_{\text{QPO}}(\pi_\theta, \pi_{\text{ref}}, -\log\sigma[\lambda(\cdot)], f', \lambda) = \tag{44}
$$
$$
\mathbb{E}_{\{y_w, y_l, x\}\sim\mathcal{D}_{\text{tr}}} -\log\sigma\left(\lambda f'\left[\frac{\pi_\theta(y_w|x)}{\pi_{\text{ref}}(y_w|x)}\right] - \lambda f'\left[\frac{\pi_\theta(y_l|x)}{\pi_{\text{ref}}(y_l|x)}\right], \lambda\right).
$$

In addition to the requirements on $f$ to form an $f$-divergence, to produce a valid $f$-DPO loss per Theorem 1 from (Wang et al., 2024a) it must be that $f'$ is invertible with $0 \notin$ domain of $f'$. Therefore the domain of $f$ will be $(0, \infty)$ and $f'(u) \to -\infty$ as $u \to 0$ because of convexity. But if this is the case, upon inspection of (44) we observe that when

$\pi_\theta(y_l|x) \to 0$, then for any fixed $\pi_\theta(y_w|x) > 0$ the input argument to the logistic function $\sigma(\cdot) = \frac{1}{1+\exp[-(\cdot)]}$ will converge to infinity, pushing the output to one and subsequently minimizing the corresponding negative-log factor. And so the global optimum can be achieved independent of the value of $\pi_{\text{ref}}$. ∎

### E.4. Derivation of (15)

$$
\begin{aligned}
d_{\text{sup}}(\pi_\theta, \pi_{\text{ref}}) &= \mathbb{E}_{\{y_1,y_2\}\sim\pi_{\text{ref}}(y|x), x\sim\mathcal{D}_x}\Big[\mathbb{KL}\big[p^*(z|y_1,y_2,x)||p_\theta(z|y_1,y_2,x)\big]\Big] \\
&= -\mathbb{E}_{\{y_1,y_2\}\sim\pi_{\text{ref}}(y|x), x\sim\mathcal{D}_x}\Big[\mathbb{E}_{z\sim p^*(z|y_1,y_2,x)}\log p_\theta(z|y_1,y_2,x)\Big] + C \\
&\equiv -\mathbb{E}_{\{y_1,y_2\}\sim\pi_{\text{ref}}(y|x), x\sim\mathcal{D}_x}\Big[p^*(z=1|y_1,y_2,x)\log p_\theta(z=1|y_1,y_2,x)\Big] \\
&\quad + -\mathbb{E}_{\{y_1,y_2\}\sim\pi_{\text{ref}}(y|x), x\sim\mathcal{D}_x}\Big[p^*(z=0|y_1,y_2,x)\log p_\theta(z=0|y_1,y_2,x)\Big], \\
&= -\mathbb{E}_{\{y_1,y_2\}\sim\pi_{\text{ref}}(y|x), x\sim\mathcal{D}_x}\Big[p^*(z=1|y_1,y_2,x)\log p_\theta(z=1|y_1,y_2,x) \\
&\qquad\qquad + p^*(z=1|y_2,y_1,x)\log p_\theta(z=1|y_2,y_1,x)\Big] \\
&= -\mathbb{E}_{\{y_w,y_l,x\}\sim\mathcal{D}_{tr}}\Big[\log p_\theta(z=1|y_w,y_l,x)\Big] \\
&= -\mathbb{E}_{\{y_w,y_l,x\}\sim\mathcal{D}_{tr}}\left[\log\left(\frac{\pi_\theta(y_w|x)}{\pi_\theta(y_w|x) + \pi_\theta(y_l|x)}\right)\right], \\
&= \mathbb{E}_{\{y_w,y_l,x\}\sim\mathcal{D}_{tr}}\left[\log\left(1 + \frac{\pi_\theta(y_l|x)}{\pi_\theta(y_w|x)}\right)\right],
\end{aligned}
\tag{45}
$$

where $C$ is a constant independent of $\theta$. Additionally, the third-to-last equality stems from the definition of how tuples $\{y_w, y_l, x\}$ are sampled. In particular, for a given pair $\{y_1, y_2\}$, by definition a proportion $p^*(z=1|y_1,y_2,x)$ of the time $y_w = y_1$, while a proportion $p^*(z=0|y_1,y_2,x) = p^*(z=1|y_2,y_1,x)$ of the time $y_w = y_2$. Hence

$$
\begin{aligned}
&p^*(z=1|y_1,y_2,x)\log p_\theta(z=1|y_1,y_2,x) + p^*(z=1|y_2,y_1,x)\log p_\theta(z=1|y_2,y_1,x) \\
&\equiv \log p_\theta(z=1|y_w,y_l,x)
\end{aligned}
\tag{46}
$$

when the latter is averaged over the preference distribution. ∎

### E.5. Further Comparisons between IPO and EXPO

Although originally designed to address certain DPO shortcomings, IPO introduces other potential sources of instability. In particular, the inherent dependency on factors $\log\big[\pi_{\text{ref}}(y_l|x)\pi_{\text{ref}}^{-1}(y_w|x)\big]$ and $(2\lambda)^{-1}$, both of which may have arbitrary magnitudes, entails that the Lipschitz constant of (9) may be arbitrarily large, destabilizing SGD. Moreover, if hypothetically $\pi_{\text{ref}}$ already closely approximates an ideal policy, minimizing (9) can actually lead to a degradation in quality (see Section 3.1).

In this regard, may be tempting to consider reparameterizing the IPO loss analogous to $\ell_{\text{EXPO}}^r$ so as to accommodate $\lambda \in [0,1]$ and reduce potential pathways for instability. But this is not a viable option for multiple reasons. First, IPO is based on reparameterizations analogous to (5) which, for a pair of responses $y_1 \neq y_2$ and the reward from (8) leads to the key equivalence

$$
\log\left[\frac{\pi_{\text{IPO}}(y_1|x)\pi_{\text{ref}}(y_2|x)}{\pi_{\text{IPO}}(y_2|x)\pi_{\text{ref}}(y_1|x)}\right] = \frac{1}{\lambda}\big[r_{\text{IPO}}(y_1,x) - r_{\text{IPO}}(y_2,x)\big].
\tag{47}
$$

The entire motivation for IPO is then to approximate this equivalence by minimizing the loss

$$
\mathbb{E}_{\{y_1,y_2\}\sim\pi_{\text{ref}}(y|x), x\sim\mathcal{D}}\left[\left(\log\left[\frac{\pi_\theta(y_1|x)\pi_{\text{ref}}(y_2|x)}{\pi_\theta(y_2|x)\pi_{\text{ref}}(y_1|x)}\right] - \frac{1}{\lambda}\big[r_{\text{IPO}}(y_1,x) - r_{\text{IPO}}(y_2,x)\big]\right)^2\right],
\tag{48}
$$

which equates to minimizing (9) per results in Azar et al. (2024). However, if we modify (9), this equivalence is compromised, i.e., we unavoidably break the foundational connection with an RLHF-like loss and implicit reward upon which IPO is entirely based in the first place.

Secondly, even if we are willing to jettison the original IPO motivation, and prefer to adopt an IPO-related hybrid loss, minimicking the weighted average of EXPO in a form such as

$$\ell(\pi_\theta, \pi_{\text{ref}}, \lambda) := \mathbb{E}_{\{y_w, y_l, x\} \sim \mathcal{D}_{\text{tr}}} \left( \log \left[ \frac{\pi_\theta(y_w|x)}{\pi_\theta(y_l|x)} \right] - \left[ \lambda \log \left[ \frac{\pi_{\text{ref}}(y_w|x)}{\pi_{\text{ref}}(y_l|x)} \right] + (1-\lambda) \log \left[ \frac{\pi^*(y_w|x)}{\pi^*(y_l|x)} \right] \right] \right)^2, \quad (49)$$

practical implementation remains a problem. In particular, as a consequence of Jensen's inequality and properties of expectations, it is no longer possible to establish the equivalent of Proposition 4.1 when using (49). Therefore an unbiased practical implementation becomes problematic, unlike the original IPO or $\ell_{\text{EXPO}}^r$. ∎

