# OpenReview forum: "Explicit Preference Optimization: No Need for an Implicit Reward Model"
_ICML.cc/2025/Conference — ICML 2025 poster_

### Official Review · Reviewer_BiLh · 2025-02-27

**Overall Recommendation:** 4

**Summary:**

The paper presents a new objective to use for preference optimization that replaces DPO-like objectives. The new objectives (EXPO) are designed to address two issues with DPO-like objectives, (1) shifting the learned policy away from the reference policy when the reference policy closely matches the optimal policy and (2) smoothly interpolating between the optimal and the reference policy where the model's behavior should more closely match that of the reference policy as lambda increases. The analysis presented a generalized Quasi Preference Optimization framing for DPO-like objectives, and then breaks down how the structure of the objective contributes to the two limitations. Then the two EXPO algorithms are introduce with one a combination of multiple objectives and the other a regression objective. An empirical study is conducted to evaluate how well EXPO aligns with the objectives of not moving the learned policy away from the reference policy when the reference policy is of a high quality and smoothly interpolating between the optimal and reference policy based on the value of lambda. The evaluations are conducted on a synthetic dataset and two real world preference datasets. The conclusions are that EXPO better exhibits the two properties than the QPO objectives, and has a higher win rate on the real world tasks.

**Claims And Evidence:**

The main claim of the paper is that the EXPO preference objective better meets the two goals of preserving the things the reference model already does well and smoothly interpolating between the reference and optimal policy that the QPO objectives. Experiments on synthetic data are used to specifically test how well EXPO meets the two goals relative to a subset of specific QPO objectives. The EXPO and QPO methods are then assessed using real data and win rate compared to the chosen responses in the corresponding dataset's test split.

Not something mentioned in the paper, but from the results it looks like the benefit of the QPO objectives is they are less sensitive to the value of the lambda hyper-parameter than the QPO objectives. This means that there are values of lambda for which the QPO results closely match of the EXPO learned models. The results in Figure 4 support this. However, the results in Figure 5 (win rate comparisons) do not account for the performance differences with respect to the value of lambda in the case of "real world" data. So it is unclear how beneficial the additions of EXPO are in real world scenarios where lambda is carefully tuned.

Following the paper and the claims requires following a lot of detailed proofs. It would be beneficial to include a higher-level intuitive explanation for the differences between the QPO and the EXPO objectives.

It is difficult to assess the claims made around Figure 8 without lines marking the optimal policy. If the lines for the optimal policy match those in Figure 3 then it looks like all methods converge to the optimal policy, not reference.

**Essential References Not Discussed:**

The correct papers appear to be discussed.

**Experimental Designs Or Analyses:**

- Given that the similarity in performance between the QPO and EXPO methods varies with the value of lambda, it would be beneficial to show how the different values of lambda impact performance on the real world tasks.
- It is not clear to me how Figure 3 addresses the interpolation goal. The results are only provided in relation to the optimal policy, but the interpolation is in relation to the reference and the optimal policies.

**Methods And Evaluation Criteria:**

It is not clear to me what exactly about the QPO objectives is problematic and exactly how EXPO addresses the issue.

In motivating the QPO generalization, it would be beneficial to explicitly spell (can be in the appendix) how IPO and DPO are expressed in terms of the QPO objective.

**Other Comments Or Suggestions:**

- It would be beneficial to either add the distance between the reference and optimal policies to the "Good Cases" plot in Figure 4 or describe where it is.

**Other Strengths And Weaknesses:**

- The paper is well written and the text is easy to follow.

**Questions For Authors:**

1. How well tuned are the lambda parameters for the Anthropic-HH and IMDB experiments? What values were used and how were they selected. This question pertains to my concerns around the QPO and EXPO methods having similar performance in Figure 4 for certain values of lambda.

**Relation To Broader Scientific Literature:**

The paper attempts to address issues faced with DPO-like objectives.

**Theoretical Claims:**

I did not. My background is more empirical than theoretical

---

> ### Author Rebuttal · Authors · 2025-03-31
>
> We are appreciative of the reviewer's helpful comments.
>
> **Comment:**
> *For the right $\lambda$, QPO can closely match EXPO ... results of Figure 4 support this ... so how beneficial is EXPO in real-world cases with tuned $\lambda$.*
>
> **Response:**
> Actually, there are no values of the QPO hyperparameter that can *simultaneously* match EXPO in both the good and bad input prompt regimes as depicted.  For example, consider IPO: for the good cases $\log \lambda > 0$ is optimal, while for the bad cases $\log \lambda \approx -1$ is optimal.  We provide another illustration of this point at this  [link](https://anonymous.4open.science/r/ICML2025_rebuttal-15908/Figure4_extention_preservation.pdf); only change is adjusting $\pi^*$. And overall, we regard reduced sensitivity to $\lambda$ as a positive even in real-world scenarios, as hyperparameter tuning can be expensive.
>
>
> **Comment:**
> *Ambiguity of Figure 8 ... it looks like all methods converge to the optimal policy, not reference.*
>
> **Response:**
> Figures 7 and 8 are meant to be viewed in tandem (and we can add explicit text to better convey their relationship).  From Figure 7, when $\lambda$ is large we observe training convergence to $\pi_{ref}$ values (0.2 and 0.4).  Meanwhile in Figure 8 we show fully converged solutions for a range of $\lambda$ values varying from small to large.  For large $\lambda$ (right side of each plot) we observe that all models produce $\pi_{ref}$ as expected.  Conversely, for small $\lambda$ (left side of each plot) we see that only EXPO maintains the optimal policy.  Again, we can easily clarify this in a revised appendix.
>
>
> **Comment:**
> *What exactly about the QPO objectives is problematic and exactly how EXPO addresses the issue ... higher-level intuitive explanation would be helpful.*
>
> **Response:**
> Good suggestion.  We agree that accessible explanations are valuable for conveying some of the subtle messages associated with our work.  In this regard, the paragraph beginning on Line 172 (righthand column) may be helpful.  There we provide a more intuitive perspective for the specific QPO cases of DPO and IPO; similar ideas hold more broadly, but are decidedly more involved to present, hence the technical aspects of our submission.  There is also another natural entry-point for understanding where QPO objectives begin to fall short.  For simplicity, consider only the special case of DPO, which is advertised as producing the minimal solution to equation (4) instantiated with an optimal reward.  Next observe how (4) behaves as $\lambda \rightarrow 0$.  Basically, once the KL term is ignored, the remaining loss will be trivially optimized by a degenerate policy assigning *all* probability mass to a single response; see equation (43) for the derivation.  In contrast, the EXPO loss is explicitly designed to reflect the *full* optimal BT policy in this same limit (not just a mode).  Appendix E.2 provides further details regarding why this distinction is important.
>
> **Comment:**
> *... it would be beneficial to explicitly spell how IPO and DPO are expressed in terms of the QPO objective.*
>
> **Response:**
> Great suggestion, and this is easy to include in a revised appendix.
>
> **Comment:**
> *Given that the similarity in performance between the QPO and EXPO methods varies with lambda, it would be beneficial to show how the different lambda values  impact performance on the real world tasks.*
>
> **Response:**
> As we have clarified in a previous response above, QPO and EXPO performance is quite distinct as $\lambda$ is varied across our synthetic experiments.  And while we do agree that further ablations with real-world data would be interesting, such testing incurs significant time and computational cost that we unfortunately cannot accommodate. That being said, in Appendix B.3 we mention $\lambda$ stability for EXPO under a limited testing range.
>
> **Comment:**
> *It is not clear to me how Figure 3 addresses the interpolation goal ...*
>
> **Response:**
> Indeed Figure 3 only shows the interpolation extreme on the optimal policy side (i.e., when $\lambda$ is small).  However, Figures 7 and 8 in Appendix B.2 complete the full picture, as we ran out of space in the main text.  For reference, there is also a pointer on Line 346 (righthand column).
>
> **Comment:**
> *Good to add the distance between the reference and optimal policies to the "Good Cases" plot in Figure 4 ...*
>
> **Response:**
> By design, the distance between $\pi_{ref}$ and $\pi^*$ is zero in Figure 4, top plot.  Please also see Line 355 (righthand side) in the text.
>
> **Comment:**
> *How well tuned is lambda for Anthropic-HH and IMDB experiments ... relationship with Figure 4.*
>
> **Response:**
> Please see our responses above regarding Figure 4, and the distinct performances of QPO vs EXPO once both good and bad cases are considered collectively.  In terms of tuning $\lambda$ on real-world data, we only explored a few distinct values because of computational cost; see also Section B.3.

---

> > ### Comment · Reviewer_BiLh · 2025-04-07
> >
> > Sorry for the delay in this message. I posted it in the wrong spot.
> >
> > The authors have addressed a number of my concerns, so I have raised by score accordingly

---

> > > ### Author Response · Authors · 2025-04-08
> > >
> > > We appreciate the reviewer's continued engagement with our paper and decision to raise the score after viewing the rebuttal.  In this regard, we notice that the score reported in Openreview has not yet changed as the discussion period is soon drawing to a close.  Given that the reviewer mentioned originally posting in the wrong spot, we were politely wondering if the reviewer may have inadvertently missed updating the score for our paper?  Thanks for your consideration.

---

### Official Review · Reviewer_tgLB · 2025-03-11

**Overall Recommendation:** 3

**Summary:**

This paper proposes a new direct preference optimization (DPO) method. The authors first formulate quasi-convex generalizations to unify some of existing DPO based methods. Then, they identify two limitations of existing DPO based methods under this formulation. One limitation is the failure to preserve optimal policies and the other one is suboptimal interpolation. To address these two limitations, the authors propose two new preference optimization objectives. Experiments on synthetic and real-world datasets validate the effectiveness of the proposed methods.
## update after rebuttal
The authors' responses address my concerns. I am leaning towards accept.

**Claims And Evidence:**

1. I am not convinced by the first limitation of existing approaches. The considered setting seems a bit synthetic to me as it only considers two types of prompts: good and bad, where $\pi^*=\pi_{ref}$ for good prompts. However, it is very likely that $\pi_{ref}$ is not optimal for all the prompts in the datasets. A more realistic setting is that the authors can prove the proposed methods can improve over $\pi_{ref}$ over all the prompts without requiring $\pi^*=\pi_{ref}$ for good prompts.

2. The synthetic evaluations support the limitation claim and the advantage of the proposed method over the existing method. However, more evidences are needed to demonstrate using the real-world dataset. For example, to address the second limitation. the authors can show the diversity of the proposed methods over existing methods while ensuring good generation quality.

**Essential References Not Discussed:**

NA

**Experimental Designs Or Analyses:**

The synthetic evaluations effectively highlight the advantages of the proposed methods in overcoming the two identified limitations. However, additional evidence is required to demonstrate these benefits on real-world datasets. For instance, to address the second limitation, the authors could showcase the diversity of the proposed methods compared to existing approaches while maintaining high-generation quality.

**Methods And Evaluation Criteria:**

The proposed methods are well motivated by the identified two limitations. However, my major concern is that the first limitation seems unreasonable to me. In practice, it is very likely that $\pi_{ref}$ is not optimal for all the prompts in the datasets. A more realistic setting is that the authors can prove the proposed methods can improve over $\pi_{ref}$ over all the prompts without requiring $\pi^*=\pi_{ref}$ for good prompts.

**Other Comments Or Suggestions:**

NA

**Other Strengths And Weaknesses:**

NA

**Questions For Authors:**

Please refer to the above questions for comments.

**Relation To Broader Scientific Literature:**

The paper is broadly related to AI alignment.

**Theoretical Claims:**

I didn't check the proof.

---

> ### Author Rebuttal · Authors · 2025-03-31
>
> We appreciate the constructive comments, and address the main points as follows (grouping where appropriate).
>
>
> **Comment:**
> *First/Main limitation: I am not convinced by the first limitation of existing approaches. The considered setting seems a bit synthetic to me as it only considers two types of prompts: good and bad, where* $\pi_{ref} = \pi^*$ *for good prompts. However, it is very likely that $\pi_{ref}$ is not optimal for all the prompts in the datasets. A more realistic setting is that the authors can prove the proposed methods can improve over over all the prompts without requiring* $\pi_{ref} = \pi^*$ *for good prompts.*
>
>
> **Response:**
> We remark that it is commonplace for theoretical results to involve simplifying assumptions, otherwise many analytical steps become infeasible.  That being said, in the present circumstance the point is not necessarily to reflect the most realistic scenario possible.  Instead, the goal is to show that even in an idealized case DPO-like methods perform below expectations, the implication being that such subpar performance is unlikely to disappear even under broader conditions.  We also reiterate that our assumption (for Theorem 3.1) is not that $\pi_{ref} = \pi^*$ for all prompts, but rather, only for so-called ideal good cases.
>
>
> **Comment:**
> *Second limitation:  The synthetic evaluations support the limitation claim and the advantage of the proposed method over the existing method. However, more evidences are needed to demonstrate using the real-world dataset. For example, to address the second limitation. the authors can show the diversity of the proposed methods over existing methods while ensuring good generation quality.*
>
> **Response:**
> As demonstrated in Figure 5 of the submission, our proposed method achieves high-quality generation on multiple real-world datasets. However, to quickly explore diversity per the reviewer's suggestion with limited time during the rebuttal period, we evaluate using the basic setup, sampling method and metrics from Section 5.3 of [this paper](https://openreview.net/forum?id=2cRzmWXK9N) (although some experimental details, like max_token are not reported in the paper; for these we adopt our settings).  Results are shown in the new table below, where generally EXPO displays higher diversity relative to DPO.  We also normalize the entropy by token length to avoid bias towards longer responses; other metrics are already implicitly normalized w.r.t. length.
>
> |     | **Normalized Entropy** $\uparrow$ | Self-Bleu $\downarrow$ | Distinct-1 $\uparrow$ | Distinct-2 $\uparrow$ |
> | --- | --- | --- | --- | --- |
> | DPO | 0.033 | 0.93 | 0.018 | 0.29 |
> | EXPO (comp) | **0.036** | 0.90 | **0.025** | **0.35** |
> | EXPO (reg) | 0.035 | **0.88** | 0.023 | 0.33 |

---

### Official Review · Reviewer_udv5 · 2025-03-18

**Overall Recommendation:** 3

**Summary:**

This paper works broadly on offline preference optimization methods, the most canonical of which is DPO, discusses a common weakness shared by all of these methods, and proposes a method that fixes this problem. The paper argues that DPO's uniform regularization to the reference policy creates problems: assume the space of prompts can be divided into two subsets, **set A**: one where the reference policy (also usually the policy one starts with during fine-tuning) is already close to optimal, and **set B**: one where the reference policy performs poorly. The paper then shows that DPO and its variants in general improves performance of the trained policy on $B$, but it inevitably comes with performance degradation on $A$. The paper then proposes their method, EXPO, that removes the reparaterization trick (designing an implicit reward function based on log probabilities), and shows that their method can improve performance on set $B$ while preserving performance on **set A**.


# Updates after Rebuttal

The authors have answered my concerns. Thanks for the additional experiments, specially the large scale experiments on Llama-3-8B.

I maintain my score at 3, and recommend acceptance of this paper.

**Claims And Evidence:**

The claims made in this paper is clear and has convincing evidence.

**Essential References Not Discussed:**

There have been prior work that has discussed the problems with DPO's implicit reward formulation. For example, [10] should be cited and discussed, but it is not. [11] should also be cited. I would request the authors to find more papers that talk about implicit reward formulations.

Also, all the papers mentioned in **Relation To Broader Scientific Literature** section above should be cited in related works, but it does not seem to be case.

**Experimental Designs Or Analyses:**

Yes, I checked the soundness of the experiments, they seem correct to me and follows what is common in relevant literature.

**Methods And Evaluation Criteria:**

The proposed methods make sense to me. I think the paper is lacking in benchmark datasets and models that it shows results on. The paper mentions multiple times that they use the same model and same datasets used in the original DPO paper. While that is great, the DPO paper is 2 years old at this point, and the same level of evaluation is no longer sufficient in my opinion. The LLM used in real-world experiments is Pythia-2.8B, which is very outdated by this point.

I would request the authors to show evidence that their method works with:

1. At least one more dataset, say UltraFeedback [1], which has ~60K prompts and (preferred, dispreferred) pairs, so it has reasonable scale.

2. At least one more LLM of 8B parameter size, say Llama-3.1-8B [2]. (It is possibly good to fine-tune the base model instead of the instruction tuned model, to show improvements resulting from EXPO).

The paper mentions that they use the same evaluation setup as the original DPO paper. But I would like to point out that the original DPO paper is around 2 years old now, so a paper submitted for review now should use more recent benchmarks/models for it to be accepted.

**Other Comments Or Suggestions:**

No other comments that I can think of.

**Other Strengths And Weaknesses:**

# Strengths

Many other recent works have also found out different problems with DPO and similar methods. This paper uncovers a major problem with DPO-like methods, shows it both theoretically and empirically. I like the paper's theoretical results.

# Weaknesses

1. The most major weakness is the scale of LLM experiments in this paper. A 2.8B old LLM does not tell practitioners enough about how much to trust this paper's results.

2. It is pretty clear from recent papers such as Deepseek [12] and other works [5, 6] that offline preference learning algorithms simply don't perform as well as their online counterparts. Another recent work such as [13] confirmed this for RLHF, where we do not have access to gold reward during training (unlike Deepseek, which trains on verifiable problems with gold reward). It is very unclear what would be the value of this paper (or in general any other paper that gives an n-th variation of an offline preference learning algorithm), given that the community might shift to more online methods like GRPO [14] anyway.

**If the authors can show that their method works with a newer model with around 8B parameter count, I would be very happy to increase this paper's score to 4.**

**Questions For Authors:**

# Questions

1. How does the memory and computational cost compare between EXPO, DPO and other key variants? It would be good to report these numbers on an 8B model against a standard dataset.

2. How does the log-probability of preferred and dispreferred responses change throughout training? Essentially I am interested in Figure 17 of [6] but for EXPO.

3. Is there any intuitive explanation of the difference between WIC and SIC conditions? I am not entirely sure if I grasp their difference and significance.

4. Does this paper's main observation, that performance improvement on set of bad prompts also result in performance decrease on set of good prompts, also happen for online preference optimization techniques? Why or why not? (**Note: this is outside the scope of this paper and the authors should not be penalized for discussing this in their paper, but I am curious and would love it if they have any explanation.**)

5. Most preference learning algorithms work on the single-turn setting. However, LLMs are being increasingly used in more and more multi-turn setting. Algorithms like DPO has a simple multi-turn extension: one can just take loss over the agent action tokens and discard the log-probabilities of the environment tokens, see [15]. Is there a way to extend EXPO to the multi-turn setting?


# References

[1] UltraFeedback: Boosting Language Models with Scaled AI Feedback, https://arxiv.org/abs/2310.01377. **Dataset link: https://huggingface.co/datasets/HuggingFaceH4/ultrafeedback_binarized**

[2] The Llama 3 Herd of Models, https://arxiv.org/abs/2407.21783.  **Model link: https://huggingface.co/meta-llama/Llama-3.1-8B**

[3] Direct Preference Optimization: Your Language Model is Secretly a Reward Model, https://arxiv.org/abs/2305.18290

[4] Scaling Laws for Reward Model Overoptimization in Direct Alignment Algorithms, https://arxiv.org/abs/2406.02900v1

[5] Is DPO Superior to PPO for LLM Alignment? A Comprehensive Study, https://arxiv.org/abs/2404.10719

[6] Preference Fine-Tuning of LLMs Should Leverage Suboptimal, On-Policy Data, https://arxiv.org/abs/2404.14367

[7] Unintentional Unalignment: Likelihood Displacement in Direct Preference Optimization, https://arxiv.org/abs/2410.08847

[8] Iterative Reasoning Preference Optimization, https://arxiv.org/abs/2404.19733

[9] The Importance of Online Data: Understanding Preference Fine-tuning via Coverage, https://arxiv.org/abs/2406.01462

[10] On the Limited Generalization Capability of the Implicit Reward Model Induced by Direct Preference Optimization, https://arxiv.org/abs/2406.01462

[11] Bootstrapping Language Models with DPO Implicit Rewards, https://arxiv.org/abs/2406.09760v2

[12] DeepSeek-R1: Incentivizing Reasoning Capability in LLMs via Reinforcement Learning, https://arxiv.org/abs/2501.12948

[13] All Roads Lead to Likelihood: The Value of Reinforcement Learning in Fine-Tuning, https://arxiv.org/abs/2503.01067

[14] DeepSeekMath: Pushing the Limits of Mathematical Reasoning in Open Language Models, https://arxiv.org/abs/2402.03300

[15] From $r$ to $Q^*$: Your Language Model is Secretly a Q-Function, https://arxiv.org/abs/2404.12358

**Relation To Broader Scientific Literature:**

The paper deals with offline preference optimization methods for fine-tuning language models, a canonical example of which is direct preference optimization or DPO [3], or more broadly other direct alignment methods [4]. The paper studies specific problems with this class of methods, some of which were discussed by these papers:

1. DPO, even the online variant, underperforming PPO due to poor regularization [5]

2. DPO often reduces the log-likelihood of the preferred response, leading to unintentional unalignment [6, 7, 8]

3. DPO's offline regularization to the reference policy can be problematic and fixed using an online way of regularization [9]

**Theoretical Claims:**

I took a brief look over the theoretical proofs, but since my expertise is not theory, I cannot comment on the correctness of the proofs. However, the results surely are interesting and relevant.

---

> ### Author Rebuttal · Authors · 2025-03-31
>
> We thank the reviewer for pointing out that our paper is clear, based on convincing evidence, and supported by interesting and relevant theory.  The reviewer also provided many constructive comments; with limited space to reply, we prioritize the main critiques as follows:
>
> **Comment:**
> *Methodology is sound, but warrants testing on larger scale models, e.g., in 8B parameter range as opposed to 2.8B range.*
>
> **Response:**
> We agree with the reviewer that the community is trending towards experimentation with larger models, particularly on less theoretically driven papers.  But even so, many recent papers still lean heavily on 2.8B-sized models or smaller for analyzing preference methods and drawing comparative conclusions between approaches.  As representative examples, the DeepSeek GRPO paper [14, Figure 5] uses a 1.3B model to compare online vs offline preference learning; likewise 1.4B-2.8B models are used for a similar purpose in [9, Table 1] and [13, Figure 3].  We also note that other more theoretical works such as the IPO paper (Azar et al.,2024) contain no real-world experiments at all and yet remain highly influential to the field.  In any event, we are not disputing the value of larger models; instead we are merely advocating that high quality contributions are still possible without them, especially when granting that compute resources and timing constraints are often limiting factors (as is the case for us within the rebuttal period).
>
> Even so, during the rebuttal window we tried fine-tuning a Llama-3.1-8B model using Lora to reduce the computational burden, and indeed EXPO is better than DPO in this limited setting. However, complete testing with further baselines or a full parameterization is unfortunately not feasible at this time.
>
> **Comment:**
> *Recent evidence suggests that online preference learning generally performs better ... so relevance of offline approaches may be waning.*
>
> **Response:**
> This is a very reasonable point to consider, and addressing it helps to further advance the relevance of our paper.  In this regard, the suggested references from the reviewer are extremely helpful.  Our overall response is three-fold:
>
> 1) The inference (from references [5,6,12,13,14] and others) that online preference learning is generally superior is to date mostly derived based on testing with QPO approaches, and usually DPO alone (or in the case of [6], an even earlier rejection-sampling approach).  Hence the open possibility remains that offline approaches that mitigate DPO deficiencies (like our EXPO) might reset the scales relative to online alternatives.  And crucially, many of the specific arguments provided for why DPO is outperformed by online approaches like PPO do *not* apply to our EXPO.  For example, in [5] it is argued that a key DPO limitation is that it does not exploit unlabeled prompt-only data, which can then lead to undesirable biases.  However, as we point on on Lines 322-324, EXPO can naturally incorporate such unlabeled prompt-only data.
>
> 2) Many references that argue in favor of online preference learning nonetheless suggest that DPO is still valuable when reformulated as an online or related on-policy iterative approach; see for example [6,9,10,11,13].  Some (e.g., [11]) even explicitly advocate for generalizing beyond DPO to online versions of IPO and the like.  Hence our analysis of QPO models in general, and EXPO in particular, remains quite relevant in a wide variety of revised online settings.
>
> 3) Even if we concede that offline preference learning is sub-optimal, in many scenarios it is still considerably simpler, possibly even with convergence guarantees.  As such, for prototyping or resource-constrained environments offline learning may enjoy some durable advantages.
>
>
> **Comment:**
> *Relation to literature, missing references ... [10] should be cited and discussed, but it is not. [11] should also be cited ...*
>
> **Response:**
> References [10] and [11] are empirically-driven studies that advocate strongly for iterative versions of DPO, e.g., exploiting DPO implicit rewards over multiple on-policy rounds.  See also our comments above that include reference to [10,11].  Overall, these works are quite complementary to our own, and we can definitely cite them in our revision.  Likewise for other references the reviewer has kindly provided.
>
> **Comment:**
> *Questions 1-5 ...*
>
> **Response:**
> 1. Memory consumption is the same as DPO.
> 2. Using unlabeled samples for (16) may help some, but further study is needed.
> 3. SIC and WIC differ strongly as $\lambda$ becomes small.  For SIC the minimal loss equates to an optimal preference distribution, while for WIC we only obtain the *mode* of an optimal distribution.  Appendix E.2 contains further details.
> 4. We have not tested this (and it could depend on the implementation), but intuition suggests that EXPO may still provide a benefit in some cases.
> 5. Multi-turn extensions of EXPO represent an interesting direction for future work.

---

> > ### Comment · Reviewer_udv5 · 2025-04-01
> >
> > Dear Authors,
> >
> > Thanks a lot for your detailed answers within the 5000 character limit.
> >
> > In general, I think the responses are satisfactory. I agree with the authors that running experiments on a 8B model might not always be feasible with a small enough model, and therefore would recommend not holding this against this paper while deciding its acceptance and rejection.
> >
> > However, it is still definitely true that certain phenomena appears on larger models, and hence it is important to see if improvements are real on an 8B model. I hope the the authors can add that to a later revision of this work.
> >
> > Other than that, I have no further comments at this moment. The paper still requires some additional experiments, so I will keep my score. Kudos to the authors however, for a cleanly written paper!

---

> > > ### Author Response · Authors · 2025-04-08
> > >
> > > We thank the reviewer for the detailed feedback and continued engagement with our rebuttal. Regarding the suggested LLaMA-3 8B (larger-scale) experiment, we can now provide additional preliminary results as feasible within time constraints. Specifically, starting from the full base model we execute SFT training via the Ultrachat-200k dataset; subsequently, we conducted preference fine-tuning with both DPO and EXPO.  For all training we use the full parameterization LLaMA-3 8B model parameters (i.e., no Lora as in our previous response).
> > >
> > > For DPO, we set $\lambda$ (referred to as $\beta$ in the DPO paper) to 0.01 and used a learning rate of 5e-7 as suggested by (Meng et al., 2024) for optimal performance. For EXPO (reg version), we simply selected $\lambda = 0.2$, the same value we used for the Anthropic HH dataset with the Pythia 2.8B model as reported in our original submission (note that this indicates some further degree of stability across scenarios).  We also maintained the same learning rate of 5e-7 for EXPO as applied to DPO. We used Alpaca-Eval2 to obtain length-controlled WinRate (a common metric for Alpaca-Eval2). Results are shown below:
> > >
> > > | DPO | EXPO (reg version)   |
> > > |-----------------------|------|
> > > | 16.7                 | 20.3 |
> > >
> > >
> > > The discrepancy between our DPO results above and those reported in (Meng et al., 2024) may be attributed to: (1) differences in batch size due to computational constraints, and (2) updates to AlpacaEval upon which evaluations depend. We hope these results help to further address reviewer scale-related comments.

---

### Official Review · Reviewer_zMiU · 2025-03-22

**Overall Recommendation:** 3

**Summary:**

This paper proposes a framework for aligning language models with human preferences without relying on implicit reward models. EXPO addresses limitations of existing methods like DPO and IPO, which suffer from suboptimal regularization and interpolation issues. The authors propose two variants: a compositional loss and a regression-based loss, which explicitly balance human preferences and reference policy adherence. Theoretical analysis shows EXPO preserves optimal policies in regions where the reference model performs well while improving elsewhere, and it satisfies strong interpolation criteria (SIC). Experiments on synthetic and real-world datasets (Anthropic HH, IMDb) demonstrate EXPO outperforms DPO, IPO, and other baselines in win rates and policy preservation.

**Claims And Evidence:**

EXPO avoids suboptimal regularization is validated by synthetic experiments showing EXPO converges to BT-optimal policies, while DPO/IPO converge to degenerate solutions.

EXPO satisfies SIC is validated by empirical results show EXPO interpolates smoothly between $\pi_{ref}$ and $\pi^*$, unlike DPO/IPO.

**Essential References Not Discussed:**

N/A

**Experimental Designs Or Analyses:**

I checked both the LLM tasks and the synthetic data tasks, and the ablation studies and did not find any issues.

**Methods And Evaluation Criteria:**

EXPO has been tested on Anthropic Helpfulness and Harmlessness (HH) preference dataset and IMDb dataset, as well as synthetic experiments with controlled preference distributions.

**Other Comments Or Suggestions:**

The same algorithm name ExPO has already been used in "Weak-to-Strong Extrapolation Expedites Alignment".

**Other Strengths And Weaknesses:**

Strengths:
Novel explicit regularization approach enhances interpretability and control.

Weaknesses:
Limited experiment results on real world tasks. Would like to see its performance on Llama-3-8B on AlpacaEval 2.

**Questions For Authors:**

Can EXPO handle non-BT preference models?

**Relation To Broader Scientific Literature:**

EXPO builds on DPO/IPO but eliminates their dependency on implicit rewards via explicit regularization. It connects to RLHF’s KL-regularization but avoids multi-stage training. The work fills a gap in ensuring policy preservation and interpolation, addressing underappreciated limitations in prior preference optimization methods.

**Theoretical Claims:**

Theorems 3.1 and 3.6 are derived, showing QPO methods (DPO/IPO) cannot preserve optimal policies or satisfy SIC. Propositions 4.2–4.3 demonstrate EXPO’s advantages. I checked the proof of Theorem 3.1.

---

> ### Author Rebuttal · Authors · 2025-03-31
>
> Thanks for checking many of the technical details of our paper, including proof and empirical materials, while acknowledging the novelty of our approach in addressing underappreciated limitations in prior preference optimization methods.  We respond to points of critique as follows.
>
> **Comment:**
> *Limited experiment results on real world tasks. Would like to see its performance on Llama-3-8B on AlpacaEval 2.*
>
> **Response:**
> During the short rebuttal window, we tried fine-tuning a Llama-3.1-8B model using Lora to reduce the computational burden, and indeed EXPO is better than DPO in this limited setting.  However, complete testing with further baselines or a full parameterization is unfortunately not feasible at this time.
>
> **Comment:**
> *The same algorithm name ExPO has already been used in "Weak-to-Strong Extrapolation Expedites Alignment".*
>
> **Response:**
> Thanks for the notice, we could easily change the name to avoid any confusion.
>
> **Comment:**
> *Can EXPO handle non-BT preference models?*
>
> **Response:**
> Good question.  In principle yes, but some theoretical results would need to be reconsidered.

---

### Decision · Program_Chairs · 2025-05-01

**Decision:**

Accept (poster)

**Comment:**

This paper introduces EXPO, an explicit preference optimization method that addresses key limitations of DPO and IPO, such as suboptimal regularization and poor interpolation behavior. The theoretical analysis is strong, and reviewers appreciated the novel framing and provable advantages. While the initial experiments were limited to smaller models, the authors provided additional results with LLaMA-3.1-8B during the rebuttal, which helped alleviate concerns. All reviewers gave weak accept scores, citing strong theoretical contributions and clear writing. The paper is therefore recommended for acceptance.